**EMBO** *reports*

# The bladder cancer m⁶A landscape is defined by global methylation dilution and focal 3′-UTR hypermethylation

Jonas Koch [1,2], Jinyun Xu [1], Felix Bormann [3], Vitor Coutinho Carneiro [1], Manuel Neuberger [4], Katja Nitschke[4], Malin Nientiedt [4,5], Philipp Erben[4], Maurice Stephan Michel[4], Manuel Rodriguez-Paredes [1] & Frank Lyko [1,5 ✉]

## Abstract

N⁶-Methyladenosine (m⁶A) is the most abundant internal modification of eukaryotic mRNAs and regulates target transcripts throughout the mRNA life cycle. Although changes in m⁶A have been reported in human cancers, technical limitations have hindered a comprehensive understanding of the cancer-associated m⁶A landscape. Here, we use GLORI-sequencing to establish the first transcriptome-wide, single-nucleotide resolution maps of m⁶A in bladder cancer. Comparing bladder cancer and healthy bladder samples, we discover two key m⁶A signatures: a global dilution of methylation and a focal hypermethylation at 3′-UTRs. The global methylation dilution results from an increased expression of unmethylated transcripts and a decreased expression of methylated transcripts. In contrast, focal 3′-UTR hypermethylation is associated with the overexpression of VIRMA, a component of the m⁶A writer complex. A functional role of VIRMA is confirmed in knockdown experiments that reveal reduced 3′-UTR methylation and oncogenic phenotypes of bladder cancer cells. Our study is the first to describe the m⁶A epitranscriptomic landscape of cancer at single-base resolution and provides first insights into the processes that generate its characteristic signatures.

**Keywords** Cancer; Epitranscriptomics; m⁶A; GLORI; VIRMA
**Subject Categories** Cancer; Chromatin, Transcription & Genomics; RNA Biology

## Introduction

N⁶-Methyladenosine (m⁶A) is the most frequent internal modification of mammalian mRNAs and is present in 0.15–0.6% of all adenosine residues (He and He, 2021). The modification is deposited in the nucleus by the m⁶A writer complex, which consists of the METTL3, METTL14, WTAP, VIRMA, ZC3H13, CBLL1, RBM15, and RBM15B proteins (Zaccara et al, 2019). METTL3 represents the catalytically active writer enzyme, with the other components of the complex being responsible for RNA binding and additional regulatory functions (Zaccara et al, 2019). While the mechanisms underlying m⁶A patterning across the transcriptome remain poorly understood, results from HeLa cells have suggested that VIRMA acts as a regulatory subunit that recruits the m⁶A writer complex to 3′-UTRs and stop codon regions (Yue et al, 2018). m⁶A is predominantly found in the DRAC(H) consensus sequence, in which D = A, G or T; R = A or G; and H = A, C or U (Dominissini et al, 2012; Linder et al, 2015). The modification is read by the proteins of the YTH family, which comprises YTHDC1, YTHDF1, YTHDF2, and YTHDF3. These proteins were also shown to mediate m⁶A-dependent downstream functions (Flamand et al, 2023; Zaccara et al, 2019). In this context, different molecular mechanisms of m⁶A-dependent transcript regulation were described, including alternative splicing, alternative polyadenylation, transport, stability, and translation (Boulias and Greer, 2023; Flamand et al, 2023; Koch and Lyko, 2024). Importantly, deregulation of m⁶A has been found to affect physiological and pathophysiological processes, including cancer development (Lan et al, 2019).

Bladder cancer is a major global health problem and the 10th most frequent cancer worldwide (Saginala et al, 2020). As 90-95% of bladder cancers originate from urothelial cells, urothelial carcinoma of the bladder (UCB) represents the most common form of bladder cancer (Dyrskjøt et al, 2023; Siegel et al, 2024). About 75% of all patients are diagnosed with non-muscle-invasive bladder cancer (NMIBC). 10–20% of NMIBC cases further progress to muscle-invasive bladder cancer (MIBC), which is characterized by low survival rates and a high metastatic potential (Berdik, 2017; Lopez-Beltran et al, 2024; Sylvester et al, 2006). Approved early detection biomarkers for UCB are currently not available (Batista et al, 2020; Gill and Perks, 2024). Also, UCB incidence and prevalence numbers are expected to increase due to population

[1]Division of Epigenetics, DKFZ-ZMBH Alliance, German Cancer Research Center, 69120 Heidelberg, Germany. [2]Faculty of Biosciences, Heidelberg University, 69120 Heidelberg, Germany. [3]Bioinformatics.Expert UG, 12305 Berlin, Germany. [4]Department of Urology and Urosurgery, Medical Faculty Mannheim, University of Heidelberg, 68167 Mannheim, Germany. [5]DKFZ Hector Cancer Institute at the University Medical Center Mannheim, Mannheim, Germany. ✉E-mail: f.lyko@dkfz.de

growth and aging (Richters et al, 2020). Therefore, novel therapeutic drug targets and biomarkers are urgently needed for improving patient prognosis.

Multiple studies reported evidence for cancer-associated changes in the m⁶A epitranscriptome (Deng et al, 2022; Deng et al, 2023). This includes aberrant expression of m⁶A regulators and changes in the m⁶A methylation of individual transcripts related to UCB development (Yang et al, 2024b). Using LC-MS/MS analysis, we have shown recently that m⁶A levels were strongly reduced in UCB when compared to matched paratumoral tissue (Koch et al, 2023). This presents an apparent contradiction to other published findings where local hypermethylation of cancer-associated transcripts was described (Cheng et al, 2019). To resolve this inconsistency and to provide detailed maps of m⁶A in cancer, novel and robust m⁶A detection techniques are required. Previous studies relied on antibody-dependent mapping approaches with limited resolution and insufficient specificity (Helm et al, 2019; Koch and Lyko, 2024; McIntyre et al, 2020). Also, they were originally designed to be qualitative rather than quantitative. However, these methods were often used in a quantitative manner without appropriate spike-in standards or improved methodologies such as m⁶A-seq2 (Dierks et al, 2021). Similarly, initial attempts to map m⁶A distribution by direct RNA-sequencing were limited by low sequencing coverage and robustness (Li et al, 2024; Zhang et al, 2024). These methodological constraints have greatly limited our understanding of cancer-associated changes in m⁶A patterning and their functional implications in cancer development and progression.

GLORI-sequencing is a newly established method that allows for the site-specific, absolute quantification of m⁶A through an unbiased chemical deamination protocol (Liu et al, 2023). As similar methods, such as whole-genome bisulfite sequencing (Lister and Ecker, 2009) have been highly successful in mapping the cancer epigenome, we adopted GLORI to generate the first transcriptome-wide, base-resolution maps of m⁶A in cancer. We sequenced and compared a set of 9 clinical UCB samples and 9 independent paratumoral control samples and uncovered systematic differences in the m⁶A landscape of UCB. More specifically, the integration of RNA expression and m⁶A methylation data revealed a global dilution of methylation in UCB tissues. Simultaneously, UCB samples showed local 3'-UTR hypermethylation, which correlated with alternative mRNA polyadenylation. Interestingly, these changes were associated with the overexpression of VIRMA, a component of the m⁶A writer complex, which we found to be clinically relevant for the progression of UCB. Our findings thus uncover key signatures of the cancer m⁶A epitranscriptome and provide first insight into the underlying mechanisms.

# Results

## Establishment of an m⁶A methylation analysis pipeline

In an initial set of experiments, we attempted to reproduce the published GLORI results from HEK293T cells (Liu et al, 2023). Therefore, we sequenced libraries from three HEK293T cell replicates, with an average yield of 200 million sequencing reads, respectively, and mapping rates of 50% (Table EV1). As the number of detectable m⁶A sites depends on both the coverage and the methylation level, we applied stringent criteria, requiring a minimum coverage of 15 independent reads and a non-conversion rate of at least 10% to call a m⁶A site. With these parameters, we obtained median A-to-G conversion ratios of >99% (Table EV2), which is similar to the reported conversion ratios and strongly limits the influence of false-positive signals (Liu et al, 2023). In total, we detected a shared 69,748 m⁶A sites in the transcriptomes of the three HEK293T cell replicates (Fig. EV1A). The majority (87%) of these sites were also detected previously (Fig. EV1B), with differences being largely due to the higher sequencing depth of the original study (Liu et al, 2023). Also, methylation levels were found to be highly reproducible (Fig. EV1C). Finally, it should be noted that our results are in excellent agreement with results that were obtained by direct RNA-sequencing of the same batch of cells (Hewel et al, 2025), which provides important orthogonal validation for our GLORI-based mapping and quantification of m⁶A sites.

In the next step, we repeated the protocol for the T24 UCB cell line and again obtained high-quality GLORI datasets (Tables EV1 and EV2). Of note, the majority of reads mapped to mRNAs (>85%), while rRNA loci only accounted for <0.01% (Table EV3), consistent with a highly efficient enrichment of mRNAs during RNA preparations. Data analysis identified more than 95,000 m⁶A sites, which were predominantly found in the DRAC(H) consensus sequence motif (Fig. 1A). Within the 15 most frequent pentanucleotide motifs, canonical DRAC(H) motifs were strongly overrepresented (Fig. 1B). Also, higher and more evenly distributed m⁶A methylation levels were detected in the DRAC(H) motifs, while the non-canonical motifs were lowly methylated (Fig. 1C). Further in-depth motif analyses showed that roughly 90% of all m⁶A sites were detected in the DRAC(H) motif, while 10% of the sites were detected in motifs with one mismatch compared to DRAC(H) (Fig. EV2A). Sites detected in motifs with more than one mismatch were almost not detected (Fig. EV2A). When investigating the motifs with one mismatch, 36% were DRACN motifs, while 64% were non-DRACN motifs (mismatch occurring in the first four bases of motif, Fig. EV2B,C). Additionally, we investigated the distribution of methylation levels in the different motifs and found that the methylation levels in the motifs unrelated to DRAC(H) were substantially lower compared to the DRAC(H) motifs (Fig. EV2D). In further analyses we investigated the effect of the METTL3 inhibitor STM2457 in T24 cells. The results showed strongly reduced methylation, both transcriptome-wide (Fig. 1D) and at the level of specific transcripts (Fig. 1E). Effect sizes progressively decreased from DRAC(H) to DRACN and non-DRACN motifs (Fig. EV2E). Based on these observations, we considered the m⁶A sites in DRAC motifs as high-confidence methylation marks, while the m⁶A sites detected in non-canonical motifs were interpreted as deamination or sequencing artefacts. We therefore restricted all further analyses to m⁶A sites detected in DRAC consensus sequence motifs. Calculating the number of m⁶A sites per transcript showed a median of $n = 4$, but several transcripts were found to have multiple (up to 154) m⁶A sites (Fig. 1F). Overall, DRAC motifs showed a bimodal methylation distribution with peaks at 20% and 95% methylation (Fig. 1G). Metagene plots showed that the majority of m⁶A sites were in the CDS and 3'-UTR of the transcripts with a characteristic peak around the stop codon (Fig. 1H). These findings demonstrate the capacity of GLORI to faithfully map m⁶A sites.

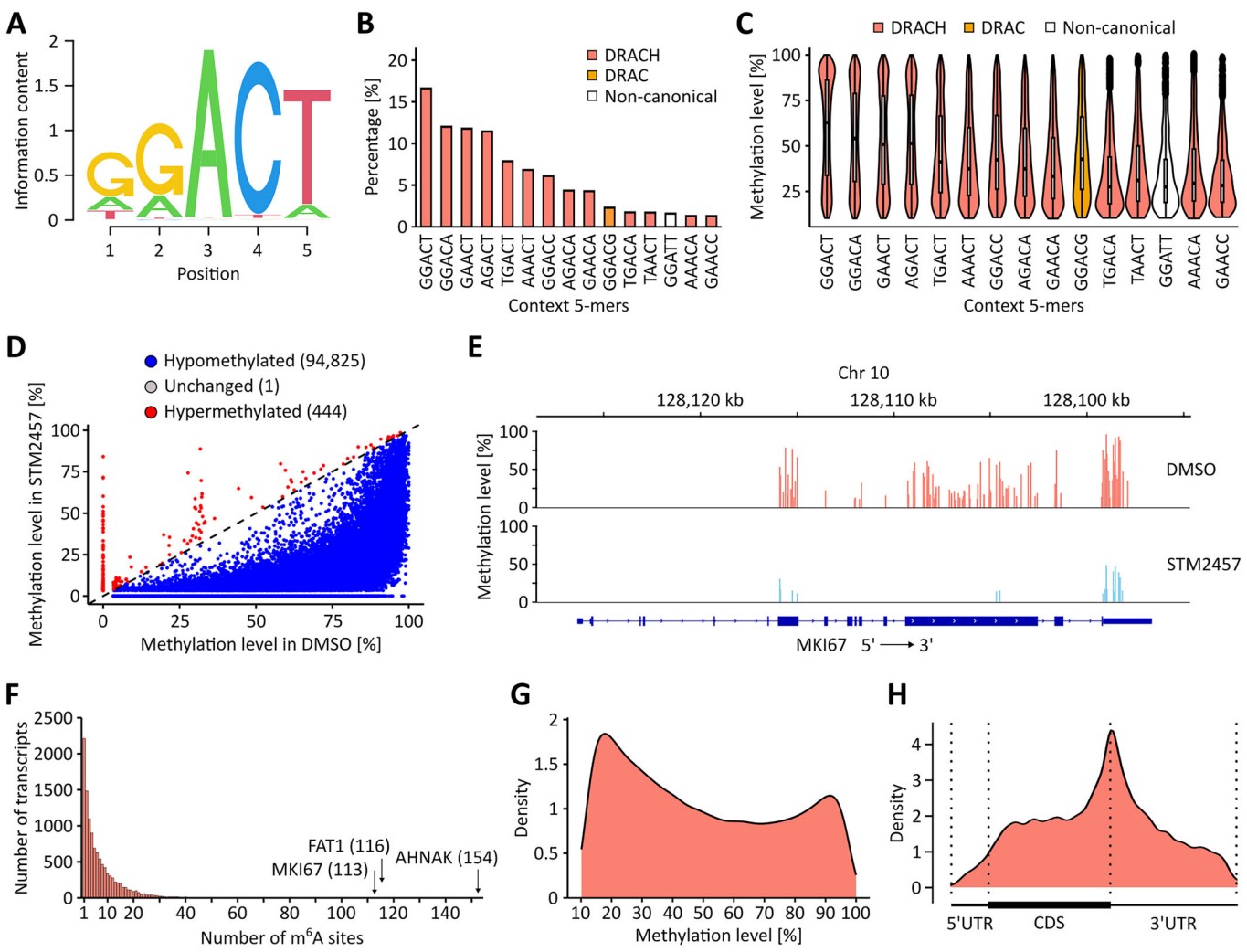

**Figure 1. GLORI-based analysis of the m⁶A landscape in T24 UCB cells.**

(A) Sequence motif analysis of m⁶A sites revealing the DRAC(H) consensus motif. (B) Frequency of the 15 most detected m⁶A site motifs. DRAC(H) motifs were detected more frequently than non-canonical motifs. (C) Quantification of the m⁶A methylation level in the 15 most detected m⁶A site motifs. The lowest m⁶A methylation levels were detected in non-canonical motifs. $n = 95,200$ m⁶A sites shared among the three T24 replicates were used for sequence motif analyses. Boxplots were generated in R using ggplot2. The centre line indicates the median (50th percentile). The box bounds represent the first and third quartiles (25th and 75th percentiles), with box height equal to the interquartile range (IQR). Whiskers extend to the most extreme values within 1.5 * IQR from the quartiles. (D) Scatter plot comparing mean m⁶A levels of methylated DRAC(H) sites between STM2457-treated and DMSO-treated T24 cells. STM2457 treatment resulted in reduced methylation levels for 94,825 sites, while 444 sites were found to have increased methylation levels. (E) IGV browser tracks showing the detected m⁶A sites in the MKI67 transcript. STM2457 treatment led to a pronounced reduction of m⁶A methylation. (F) Quantification of m⁶A sites per transcript. While the median number was $n = 4$, some transcripts harbored very high numbers of m⁶A sites, as indicated. (G) Kernel density plot showing that the majority of m⁶A sites have either low or high methylation levels. (H) Metaplot showing that detected m⁶A sites predominantly occur in the CDS and 3'-UTR regions with a peak density surrounding the stop codon. These analyses were performed based on $n = 3$ biological replicates.

## Comparative analysis of UCB and control samples

In the next step, we applied our GLORI pipeline to nine UCB and nine independent paratumoral control samples. Clinical information about the patient samples is provided in Table EV4. The quality of the corresponding GLORI datasets matched the high standards obtained with the HEK293T and T24 cell lines (Tables EV1 and EV2). Initial sequence motif analyses confirmed methylation in the consensus DRAC(H) motif for both sample groups (Fig. 2A) and did not reveal any detectable differences in pentanucleotide motif frequencies and methylation levels

(Fig. 2B,C). These results strongly suggest that m⁶A motif specificity is retained in UCB. In total, approximately 42,000 detected m⁶A sites were shared among the UCB samples, while roughly 48,000 m⁶A sites were shared among the control samples and used for further downstream analysis. To assess whether differences in m⁶A site detection across samples could be explained by differences in coverage or conversion, we compared coverage and methylation levels between shared and non-shared sites. Indeed, shared sites that were detected in all 9 samples of a group showed higher median coverage (UCB: 129 vs. 37.7 reads; Control: 87.3 vs. 31.3 reads, Fig. EV3A,C) and

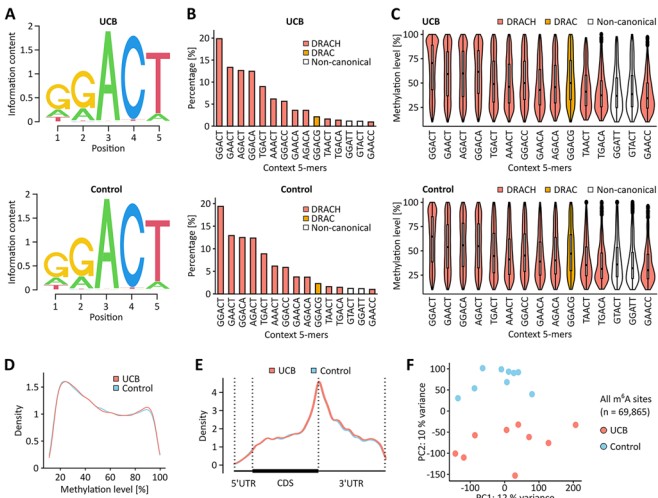

**Figure 2. The m⁶A epitranscriptomic landscapes of UCB and non-malignant uroepithelial tissue show moderate, but systematic differences.**

(A) Sequence motif analyses using detected m⁶A sites in UCB and control tissue samples. In both tissue types, m⁶A sites were primarily detected in the DRAC(H) consensus sequence motif. (B) Frequencies of the 15 most detected motifs. (C) Quantification of the methylation level in the 15 most detected motifs. $n = 42{,}339$ m⁶A sites shared among the nine UCB tissue replicates were used for sequence motif analyses. $n = 48{,}061$ m⁶A sites shared among the nine control tissue replicates were used for sequence motif analyses. Boxplots were generated in R using ggplot2. The centre line indicates the median (50th percentile). The box bounds represent the first and third quartiles (25th and 75th percentiles), with box height equal to the IQR. Whiskers extend to the most extreme values within 1.5 * IQR from the quartiles. (D) Kernel density plot showing the distribution of m⁶A site methylation levels in the UCB and control datasets. (E) Metagene plots demonstrating that the detected m⁶A sites predominantly occur in CDS and 3'-UTR regions independent of the sample group. (F) Principal component analysis demonstrating that UCB and control tissue samples can be separated based on their m⁶A signatures. These analyses were performed based on $n = 9$ biological replicates.

higher median methylation levels (UCB: 0.495 vs. 0.278; Control: 0.491 vs. 0.273, Fig. EV3B,D) compared to sites detected in fewer samples. These findings indicate that strong signals (i.e., a combination of high coverage and high methylation) are more likely to be shared across samples. When comparing the distribution of methylation levels (Fig. 2D) and the localization of these m⁶A sites (Fig. 2E), we could not identify any major differences between UCB and control samples. Interestingly, however, principal component analysis based on all m⁶A sites showed that the tumor samples could be clearly separated from the controls (Figs. 2F and EV4A). These findings strongly suggest the presence of cancer-specific signatures in the m⁶A epitranscriptomic landscape.

## Differential methylation of cancer-related transcripts

In the next step, we sought to further investigate the cancer-associated m⁶A pattern changes by identifying differentially methylated m⁶A sites using stringent criteria ($|\Delta m^6A| \geq 10\%$ and $p < 0.05$). The 10% cutoff was determined by analyzing the standard deviations (SD) of the methylation levels of each DRAC site in both groups. The vast majority of sites showed a SD < 10% in both tissue groups, with a mean of approx. 1% and the 90th

percentile at <6% (Table EV5). A methylation level difference of 10% therefore robustly exceeds the observed (stochastic) variability. Transcripts were classified as hypermethylated or hypomethylated if at least one m⁶A site showed increased or decreased methylation, respectively. Transcripts with changes in both directions were 'labeled hyper- and hypomethylated'. We identified 1921 sites with increased methylation levels in cancer (hypermethylated) and 1238 sites with reduced methylation levels in cancer (hypomethylated, Fig. 3A). Principal component and additional dimensionality reduction analyses based on these 3159 differentially methylated sites again showed a clear separation of the tumor and control sample groups (Figs. 3B and EV4B). On the transcript level, 1186 transcripts were identified to be hypermethylated, while 902 transcripts were hypomethylated (Fig. 3C). Only a small fraction of transcripts had both hyper- and hypomethylated m⁶A sites, indicating that most m⁶A sites change consistently across transcripts (Fig. 3C). Pathway analyses showed that the differentially methylated transcripts were enriched in several pathways that have been linked to UCB development (Goriki et al, 2018; Knowles and Hurst, 2015; Sui et al, 2017), including TNFα, NOTCH, p53, and TGFβ signaling, as well as pathways related to apoptosis and epithelial mesenchymal transition (EMT, Fig. 3D). Additional differentially methylated transcripts included MYC, SMAD3, and BTG2, which have all been linked to UCB (Cheng et al, 2019; Mao et al, 2015; Millet and Zhang, 2007; Yuniati et al, 2019) and were found hypermethylated in their CDS and 3'-UTR regions (Figs. 3E and EV5).

## Cancer-associated hypomethylation is due to cancer-associated changes in transcript abundance

As we observed both hypomethylation and hypermethylation in cancer samples, we performed further analyses to define the signatures of the cancer-associated m⁶A epitranscriptome. To address cancer-related differences in transcript abundance, we performed RNA-sequencing on the same tissue samples that we had used for GLORI. Principal component analysis showed that both sample groups could be separated based on their gene expression profiles (Fig. EV6A). Differential gene expression analysis demonstrated pronounced global deregulation of gene expression in UCB when compared to the control tissue, with 6,091 differentially expressed genes (Fig. EV6B). Cumulative analysis of normalized transcript levels showed that few transcripts made up a high proportion of the total transcriptome (Fig. 4A), with roughly 100 transcripts making up 50% of the transcriptome in both groups. This raised the possibility that methylation and/or expression changes in these highly abundant transcripts could strongly impact the global methylation level. Indeed, integrated analysis of GLORI- and RNA-sequencing data showed a significant global hypomethylation of UCB tissue samples (Fig. 4B), which confirmed the results observed in our previous LC-MS/MS analyses (Koch et al, 2023). Further data analysis revealed that unmethylated, highly abundant transcripts were upregulated, while highly methylated, highly abundant transcripts were downregulated in UCB (Fig. 4C). For example. several highly methylated transcripts (aggregate methylation >2), including EGR1, JUN, JUNB, and FOS, were markedly downregulated

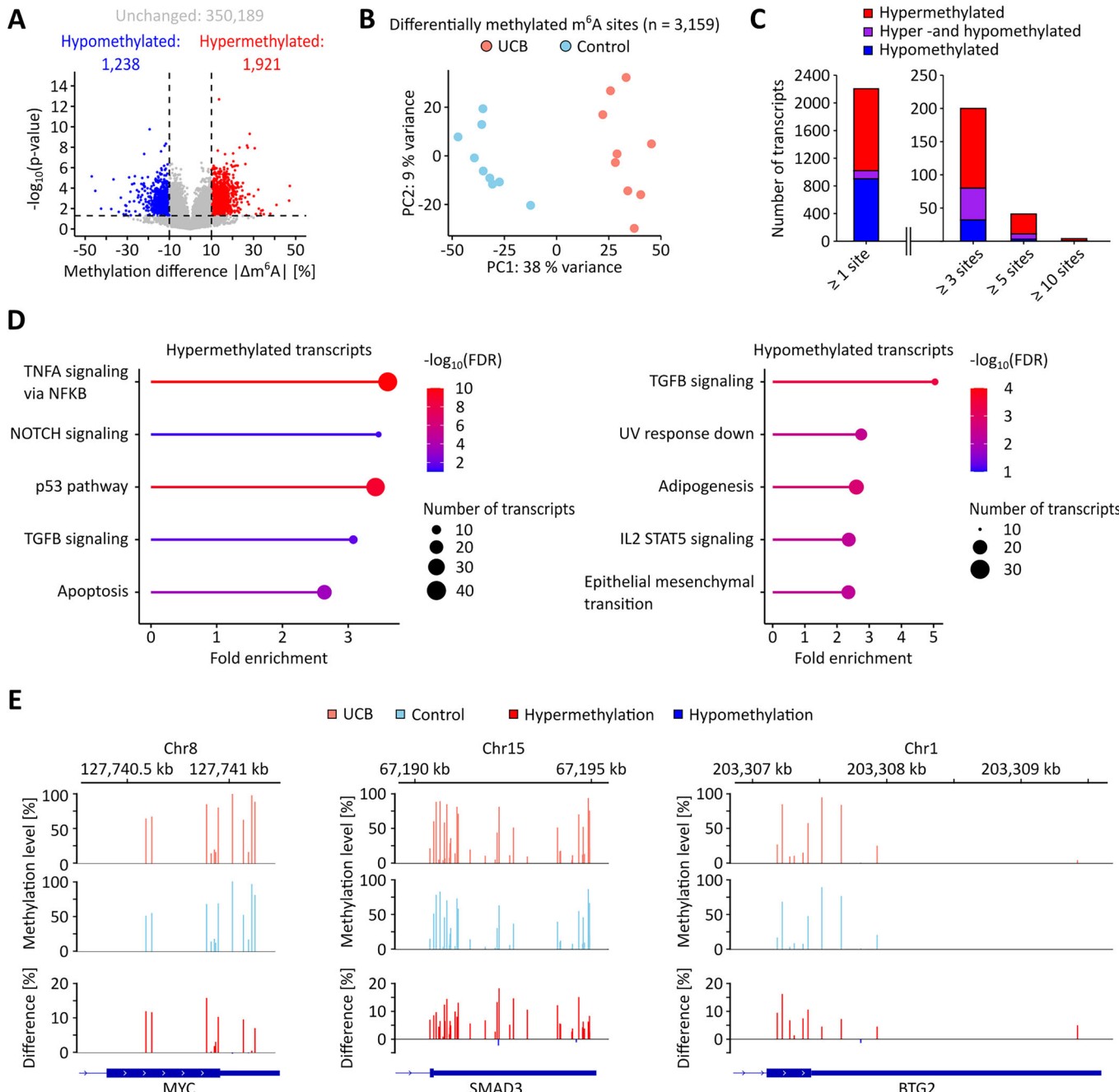

**Figure 3. The UCB m⁶A epitranscriptome is characterized by both hypo- and hypermethylation.**

(A) Differential methylation analysis revealed that 1921 m⁶A sites were hypermethylated, while 1238 m⁶A sites were hypomethylated. Thresholds: Absolute difference in methylation level >10%, $p < 0.05$. $n = 353,348$ DRAC(H) sites were analyzed for differential methylation comparing nine UCB and nine control tissue samples. Statistical significance was assessed using beta binomial models. Unmethylated DRAC(H) sites are included in this analysis. (B) PCA showing that patient samples can be separated based on the differentially methylated m⁶A sites. (C) On the transcript level, 1186 transcripts were found to be hypermethylated, 117 transcripts had both hyper- and hypomethylated m⁶A sites, and 902 transcripts were hypomethylated, indicating that the majority of transcripts are coordinately hyper- or hypomethylated. (D) Top 5 enriched pathways based on differentially methylated transcripts. (E) Prominent examples for hypermethylated transcripts in UCB. These analyses were performed based on $n = 9$ biological replicates.

in UCB (Fig. 4D). In contrast, several unmethylated transcripts from the S100 family were upregulated in UCB (Fig. 4E) These findings strongly suggest that m⁶A marks become "diluted" in cancer samples, due to the increased presence of unmethylated transcripts.

## Cancer-associated hypermethylation is enriched at 3′-UTRs and associated with increased VIRMA expression

To further characterize the contrasting signature, i.e., cancer-associated hypermethylation, we performed sequence motif

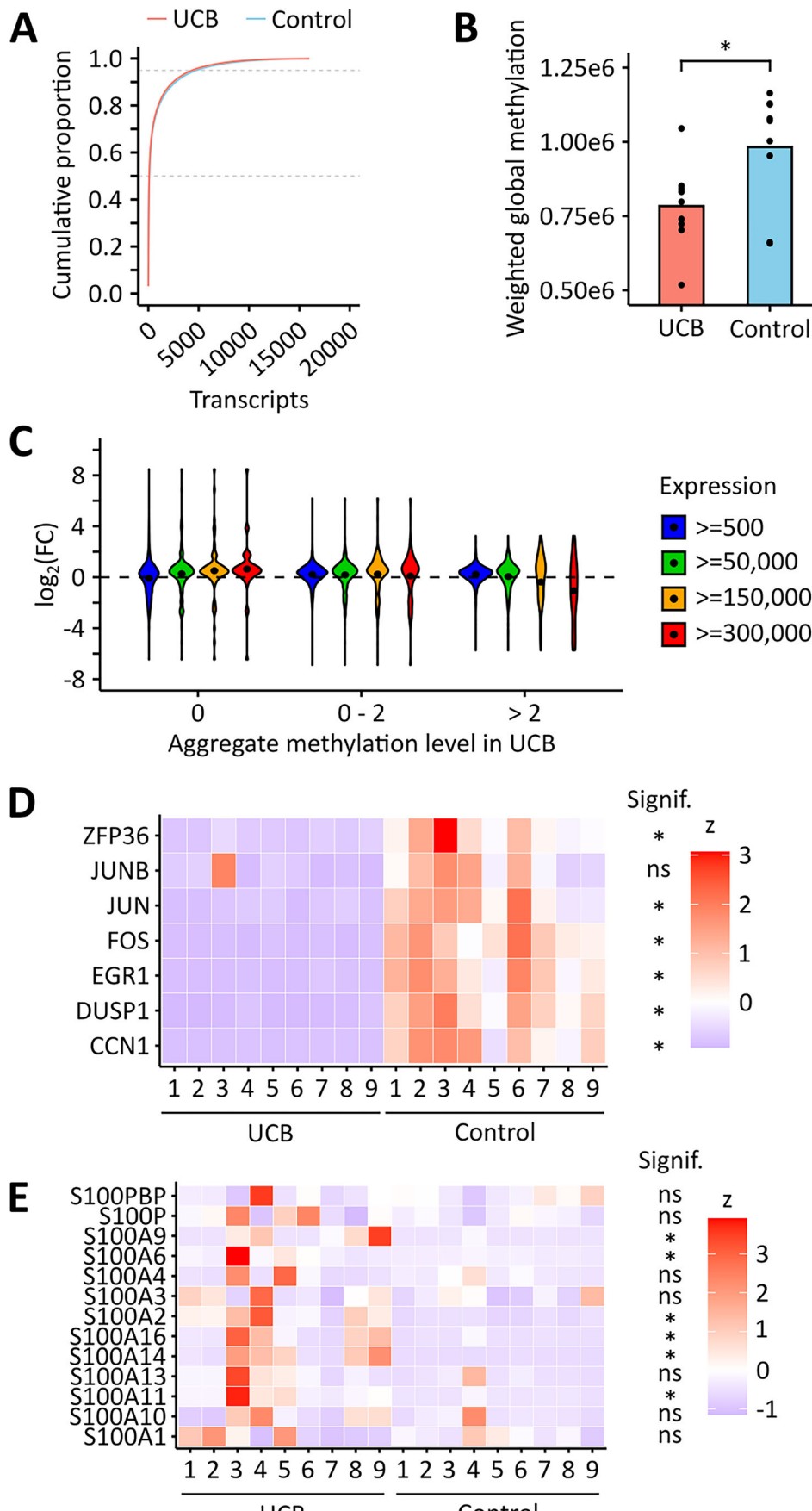

**Figure 4.   Global m⁶A hypomethylation in UCB results from changes in transcript abundance.**

(A) Cumulative proportion plot of control and UCB tissues. Few transcripts make up a high proportion of the transcriptome in both conditions. (B) Weighted global methylation levels of control and UCB tissues. The analysis confirms a global hypomethylation in UCB. *$p = 0.026$, t-test. (C) Global analysis of transcript expression changes in UCB considering the methylation status of the transcripts. When analyzing all transcripts, no major trends were observed. When restricting the analysis to highly abundant transcripts, an upregulation of unmethylated transcripts as well as a downregulation of highly methylated transcripts was observed in UCB. Expression and methylation data from $n = 13,269$ transcripts was used for this analysis. (D) Heatmap showing selected transcripts that had highest methylation levels in the lists of the most abundant transcripts in control and UCB tissues. Six out of the seven transcripts were downregulated in UCB. Threshold: q < 0.05. (E) Heatmap showing the expression of the S100 transcripts family. Several transcripts were found to be upregulated in UCB. Threshold: q < 0.05. These analyses were performed based on $n = 9$ biological replicates.

analyses. The results showed that hypermethylation was frequently detected in the GGACT motif, while hypomethylated m⁶A sites were often found in the TGACT and AAACT motifs (Fig. 5A). Furthermore, metagene analyses found that hypermethylated m⁶A sites were predominantly located in the 3'-UTR, especially in proximity to the stop codon, while no clear patterns were detectable for hypomethylated sites (Fig. 5B). As m⁶A methylation changes have been shown to affect mRNA polyadenylation (Yue et al, 2018), we analyzed our datasets for evidence of alternative polyadenylation. Indeed, the majority of transcripts had negative difference of Percentage of Distal polyA site Usage Indices (PDUIs) indicating that they were shortened (Fig. 5C). To test whether m⁶A could affect alternative polyadenylation in UCB, we expanded our alternative polyadenylation analyses to UCB METTL3 knockout cell clones and to UCB cell lines treated with the METTL3 inhibitor STM2457. RNA-seq data analysis showed that most transcripts had positive ΔPDUIs upon METTL3 knockout and inhibition (Figs. 5D and EV7). The comparison of the proportion of m⁶A-methylated transcripts that are 3'-UTR lengthened or shortened also showed that shortened transcripts are more likely to be m⁶A-methylated (Fig. 5E). Our results thus establish hypermethylation near stop codons as a second important signature of the cancer-associated m⁶A epitranscriptome and indicate that it may affect alternative polyadenylation.

Hypermethylation of m⁶A sites in 3'-UTR and stop codon regions has been associated with VIRMA in HeLa cells (Yue et al, 2018). We therefore determined the mRNA expression levels of VIRMA in our sample set. The results showed that VIRMA was significantly upregulated in UCB (Fig. 5F). Combined TCGA and GTEx cohort analyses further confirmed that VIRMA is significantly overexpressed in UCB when compared to healthy bladder tissue (Fig. 5G). To explore the underlying mechanism driving VIRMA overexpression, we examined genetic alterations in UCB patients. Notably, VIRMA gene amplification was observed in approximately 6% of tumors (Fig. 5H), which is consistent with the known amplification rates of other UCB-associated genes, including MYC (2.9–3.3% (Kluth et al, 2023; Zaharieva et al, 2005)), FGFR1 (3.7–7% (Bou Zerdan et al, 2023; Helsten et al, 2016)), and HER2 (8–8.7% (Bou Zerdan et al, 2023; Fleischmann et al, 2011)). The link between VIRMA amplification and overexpression was further supported by a correlation analysis of VIRMA copy number values and mRNA expression levels, which demonstrated a highly significant ($p = 3.02e{-}60$, Pearson correlation) positive association (Fig. 5I). Finally, survival analysis revealed that UCB patients with higher VIRMA expression had a significantly worse prognosis (Fig. 5J). These findings suggest that VIRMA can play an important

role in UCB progression, through increased expression and hypermethylation of 3'-UTRs.

## VIRMA depletion reduces m⁶A methylation and oncogenic phenotypes of UCB cells

To further analyze the functional relevance of VIRMA in UCB, we used shRNAs to knock down VIRMA expression. This resulted in a pronounced reduction of VIRMA mRNA and protein expression in UM-UC-3 cells (Fig. 6A) and are more moderate reduction in RT4 cells (Fig. EV8A). Subsequent GLORI-seq analysis revealed a marked global reduction in m⁶A levels in both cell lines when comparing the mean methylation levels of methylated DRAC(H) sites across all replicates (Fig. 6B and Fig. EV8B). To identify VIRMA-dependent m⁶A sites, we performed differential methylation analysis ($|\Delta m⁶A| \geq 10\%$ and $p < 0.05$), which revealed widespread hypomethylation upon VIRMA KD (Figs. 6C and EV8C). Notably, when examining the positional distribution of these hypomethylation events, we observed that the strongest and densest methylation level reductions localized to regions surrounding the stop codon (Figs. 6D and EV8D), consistent with a role of VIRMA in 3'-UTR methylation. Global alternative polyadenylation profiling in the VIRMA KD models also showed an effect on transcript lengthening, which appeared pronounced in UM-UC-3 VIRMA KD cells (Fig. 6E) and more moderate in RT4 VIRMA KD cells (Fig. EV8E). The observed differences may be related to differences in KD efficiency, which was more pronounced in UM-UC-3 cells. At the individual transcript level, selected target transcripts showed both a loss of m⁶A methylation within terminal exons and 3'-UTRs and preferential distal polyadenylation site usage, while control transcripts did not display a consistent trend in either direction (Fig. EV9). Notably, m⁶A-dependent regulation of AFF4 and ITGA6 has previously been described in bladder cancer (Cheng et al, 2019; Jin et al, 2019), while the remaining target genes are known cancer-associated transcripts reported to be m⁶A-regulated in other tumor entities (Cai et al, 2021; Hirayama et al, 2020; Sang et al, 2022; Wang et al, 2023; Zhao et al, 2023). Control genes lack reported roles of m⁶A-dependent regulation in cancer. Finally, we also analyzed the effect of VIRMA KD on cancer cell phenotypes. The results showed that VIRMA KD strongly impaired cell proliferation and colony-forming capacity, accompanied by elevated Caspase-3/7 activity, indicating increased apoptotic signaling in both cell lines (Figs. 6F–H and EV8F–H). These findings are consistent with our observations in UCB patients and support a functional role for VIRMA in promoting tumor cell growth and survival.

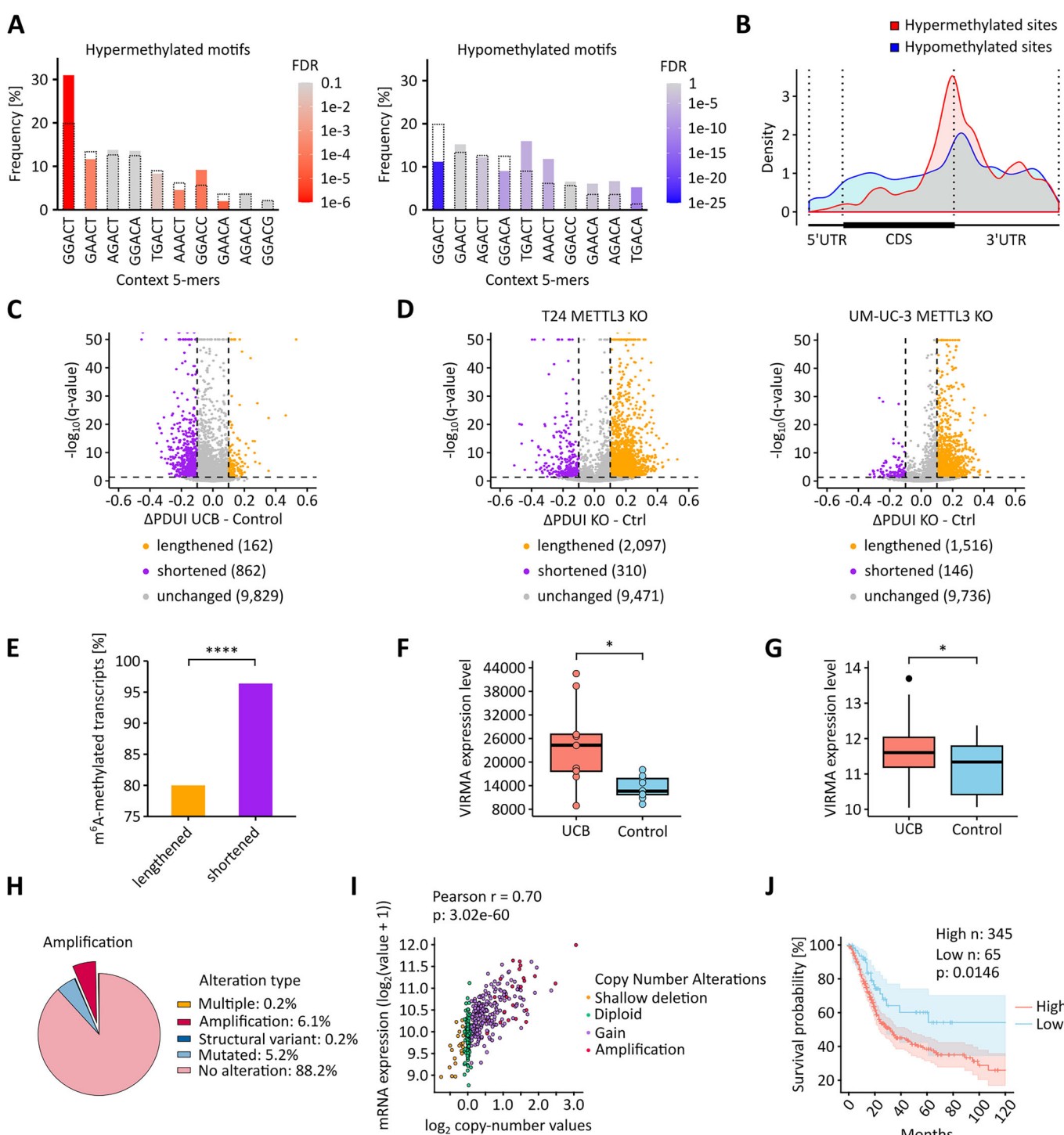

## Discussion

We have used GLORI-sequencing of nine UCB tumor samples and nine independent paratumoral control samples to establish the first base-resolution maps of the m⁶A in UCB. Sequence motif analyses showed that GLORI detected m⁶A sites predominantly in DRAC(H) consensus sequence motifs, consistent with the known specificity of the m⁶A writer complex (Dominissini et al, 2012;

Linder et al, 2015; Meyer et al, 2012). While cancer-related mutations in the m⁶A writer complex have been suggested to induce m⁶A methylation in non-canonical motifs (Zhang et al, 2023), our GLORI analysis of UCB samples showed that m⁶A sites were predominantly detected in the context of the DRAC(H) motif. These findings support earlier studies reporting relatively robust m⁶A profiles across different physiological cell and tissue types (Liu et al, 2020; Schwartz et al, 2014), and expand them to cancer.

**Figure 5.   Local 3'-UTR hypermethylation is associated with an upregulation of VIRMA in UCB.**

(A) Frequency of the motifs detected in hypermethylated and hypomethylated m6A sites, respectively. Bars with dashed outlines represent the overall frequency of the respective motif in the cancer samples. Colored bars represent the frequency of the respective motif among the differentially methylated m6A sites. (B) Metagene plot showing that hypermethylated m6A sites predominantly occur in regions surrounding the stop codon. (C) Detection of alternative polyadenylation events using DaPars. Thresholds: Absolute difference in PDUI > 0.1, FDR < 0.05. The majority of transcripts have a negative ΔPDUI in UCB. $n = 10,853$ dynamic alternative polyadenylation usages were identified comparing nine UCB and nine control tissue samples. (D) Detection of alternative polyadenylation events in UCB METTL3 knockout clones. Global reduction of m6A leads to lengthening of transcripts. $n = 11,878$ and $n = 11,398$ dynamic alternative polyadenylation usages were identified comparing three METTL3 knockout (KO) and three control cell line samples. (E) Comparison of the proportion of m6A methylation in 3'-UTR lengthening or shortening. ****$p = 6.301e-10$, Fisher's exact test. (F) Comparison of VIRMA mRNA expression between nine UCB and nine control tissue samples in our dataset. *$p = 0.011$, Mann–Whitney-U test. (G) Comparison of VIRMA mRNA expression between control ($n = 28$) and UCB ($n = 407$) tissue samples. Cancer samples are from the TCGA-BLCA cohort, control samples were combined from the TCGA-BLCA and GTEx cohorts. *$p = 0.016$, Mann–Whitney-U test. Boxplots were generated in R using ggplot2. The centre line indicates the median (50th percentile). The box bounds represent the first and third quartiles (25th and 75th percentiles), with box height equal to the IQR. Whiskers extend to the most extreme values within 1.5 * IQR from the quartiles. (H) Overview of genetic alterations of VIRMA in the TCGA-BLCA dataset. (I) Scatter plot depicting the correlation between the $\log_2$ copy-number values and VIRMA mRNA expression levels. (J) 10-year overall survival analysis of the TCGA-BLCA cohort. Patients were stratified into VIRMA-high ($n = 345$) and VIRMA-low ($n = 65$) groups. $p = 0.0146$, log-rank test.

However, we also identified systematic alterations between the m6A profiles of UCB and control tissues, which allowed their clear and unambiguous separation in dimensionality reduction analyses. A prominent example was the MYC oncogenic transcript, which is known to be regulated in an m6A-dependent manner in UCB (Cheng et al, 2019), and which we found to be hypermethylated in its CDS and 3'-UTR regions. Furthermore, we found pronounced hypermethylation in transcripts from other cancer genes, such as SMAD3 and BTG2 (Mao et al, 2015; Millet and Zhang, 2007; Yuniati et al, 2019). These findings suggest the possibility to develop novel biomarkers from the m6A profile of UCB, which would address a major unmet clinical need for this tumor entity (Batista et al, 2020).

By integrating information about mRNA abundances from standard RNA-sequencing into our GLORI analyses, we determined methylation levels relative to transcript abundance. This novel computational approach allows for a more accurate and comprehensive assessment of the m6A epitranscriptomic landscape by avoiding bias from highly abundant or underrepresented transcripts. When comparing the correspondingly adjusted methylation levels of UCB and control tissues, we found a global reduction of m6A in UCB, which is consistent with our previous data obtained by LC-MS/MS (Koch et al, 2023). Further analyses suggested that this global reduction was primarily caused by the upregulation of abundant, unmethylated transcripts, which diluted the global methylation level, and by the downregulation of abundant, highly methylated transcripts. These observations are consistent with the hypothesis that changes in m6A levels across different subcellular compartments, treatments, or tissue samples result from changes in the mRNA metabolism that affect transcript abundance (Shachar et al, 2024). Furthermore, our results highlight the limits of standard m6A quantification and/or mapping approaches that do not consider transcript abundances.

Simultaneously, our analysis identified a high density of hypermethylated sites in UCB near stop codons, a feature that has been linked to VIRMA and alternative polyadenylation (Yue et al, 2018). The depletion of VIRMA resulted in a pronounced global loss of m6A, in line with LC-MS/MS measurements from four VIRMA-depleted breast cancer cell lines (Lee et al, 2023). The strongest reduction was shown to occur at 3'-UTR sites, consistent with the existing model in which VIRMA guides the writer complex toward these regions (Yue et al, 2018).

Alternative polyadenylation is the process by which different isoforms of the same transcript can be formed based on the usage of proximal or distant polyadenylation sites, which can affect transcript stability, localization, and translation (Tian and Manley, 2017), with implications for tumor formation (Yuan et al, 2021). When analyzing alternative polyadenylation events in our samples, we found pronounced 3'-UTR shortening of transcripts in UCB, which is consistent with previous findings in UCB and other cancer entities (Xia et al, 2014). Conversely, VIRMA depletion promoted 3'-UTR lengthening, which supports published data describing a link between m6A and 3'-UTR shortening (Molinie et al, 2016; Yue et al, 2018). Furthermore, RNA-dependent interactions between VIRMA and the polyadenylation factors CPSF5 and CPSF6 have been reported (Yue et al, 2018). Together, these findings raise the possibility that m6A deposition by the writer complex and VIRMA cooperate with the polyadenylation machinery to promote proximal polyadenylation site selection. Because widespread transcript shortening is a common feature of cancer transcriptomes, often enabling escape from post-transcriptional repression (Xia et al, 2014), the frequent upregulation of METTL3 and VIRMA in tumors (Destefanis et al, 2024; Koch and Lyko, 2024) may contribute to oncogenic 3'-UTR remodeling.

Our findings also show that VIRMA KD reduced cell proliferation as well as colony formation, and enhanced apoptosis signaling in two UCB cell lines, demonstrating that VIRMA promotes an oncogenic cellular phenotype. Similar results have also been described in breast cancer (Lee et al, 2023; Li et al, 2023), pancreatic cancer (Yang et al, 2024a), head and neck squamous cell carcinoma (Zhu et al, 2024), and nasopharyngeal carcinoma (Zheng et al, 2023). Furthermore, VIRMA has been reported to be upregulated and amplified across many cancer entities, and its elevated expression has been associated with poor patient prognosis (Destefanis et al, 2024; Zhu et al, 2021). Our findings are in agreement with these studies and provide important mechanistic support for a functional role of VIRMA in UCB progression through m6A deposition.

However, the precise molecular mechanism underlying the association between m6A, its regulators, and polyadenylation factors remains unresolved, also given the inconsistent findings that were reported previously (Ke et al, 2015; Molinie et al, 2016; Ries et al, 2023; Yue et al, 2018). For example, it is currently unclear whether m6A directly modulates the recruitment or stability of

polyadenylation factors or alters RNA structure to expose specific polyadenylation sites. This could be addressed by combining VIRMA or METTL3 interference with CLIP-seq of core polyadenylation factors and structure-probing approaches, like SHAPE-MaP (Siegfried et al, 2014) around (alternative) polyadenylation sites. The impact of VIRMA-dependent m⁶A on alternative polyadenylation could be investigated by integrating VIRMA and METTL3 perturbation models with 3′-end-targeted sequencing methods (Yu et al, 2020; Zheng et al, 2016), and/or reporter assays with mutated 3′-UTR m⁶A sites. Furthermore, a direct transcript-level assessment of how m⁶A methylation influences alternative polyadenylation would require robust multi-modal single-molecule sequencing approaches capable of simultaneously resolving m⁶A modifications and mRNA 3′-end usage. Current long-read sequencing technologies, including nanopore sequencing, are not yet ideally suited for this purpose due to their limited accuracy in m⁶A basecalling.

Altogether, our study identifies two separate signatures and mechanisms that alter the m⁶A epitranscriptomic landscape in cancer (Fig. 7). Global m⁶A methylation levels become diluted due to the upregulation of unmethylated transcripts and the downregulation of methylated transcripts, while focal m⁶A hypermethylation in 3'-UTR regions is associated with overexpression of VIRMA. The highly precise and quantitative results generated by GLORI were critical for uncovering these signatures. Further work will be needed to elucidate the potential of these signatures for identifying biomarkers for early detection of bladder cancer.

# Methods

### Reagents and tools table

| Reagent/Resource | Reference or Source | Identifier or Catalog Number |
|---|---|---|
| **Experimental Models** | | |
| HEK293T (*Homo sapiens*) | European Collection of Authenticated Cell Cultures (ECACC) | RRID: CVCL_0063 |
| RT4 (*Homo sapiens*) | ECACC | RRID: CVCL_0036 |
| T24 (*Homo sapiens*) | ECACC | RRID: CVCL_0554 |
| UM-UC-3 (*Homo sapiens*) | ECACC | RRID: CVCL_1783 |
| Human bladder tumor and paratumoral patient samples | Department of Urology and Urosurgery, Medical Faculty Mannheim | 2015-549N-MA |
| **Recombinant DNA** | | |
| psPAX | Addgene | Cat #12260 |
| pMD2.G | Addgene | Cat #87360 |
| pLentiCRISPR v2 | Addgene | Cat #52961 |
| **Antibodies** | | |
| β-Actin monoclonal antibody | Sigma-Aldrich | Cat A5316 |
| VIRMA polyclonal antibody | Proteintech | Cat 25712-1-AP |

| Reagent/Resource | Reference or Source | Identifier or Catalog Number |
|---|---|---|
| Donkey anti-rabbit IgG-HRP | Santa Cruz Biotechnology | Cat sc-2313 |
| Donkey anti-mouse IgG-HRP | Thermo Fisher Scientific | Cat A16011 |
| **Oligonucleotides and other sequence-based reagents** | | |
| Anti-METTL3 sgRNA and anti-VIRMA shRNA sequences | This study and Horizon Discovery | Table EV6 |
| **Chemicals, Enzymes and other reagents** | | |
| RNAlater solution | Thermo Fisher Scientific | Cat AM7021 |
| DMEM, high glucose, pyruvate medium | Gibco | Cat 41966052 |
| McCoy's 5A (Modified) medium | Gibco | Cat 26600023 |
| Fetal bovine serum (FBS) | Gibco | Cat 10500064 |
| Penicillin-Streptomycin (P/S) | Gibco | Cat 15140-122 |
| TRIzol | Thermo Fisher Scientific | Cat 15596026 |
| MEGAclear Transcription Clean-Up Kit | Thermo Fisher Scientific | Cat AM1908 |
| Dynabeads mRNA Purification Kit | Thermo Fisher Scientific | Cat 61006 |
| NEBNext Magnesium RNA Fragmentation Module | New England Biolabs | Cat E6150S |
| RNA Clean & Concentrator-5 kit (DNase I included) | Zymo Research | Cat R1013 |
| Glyoxal solution | Sigma-Aldrich | Cat 50649 |
| DMSO | AppliChem | Cat A3672,0250 |
| Boric acid | Sigma-Aldrich | Cat B0394 |
| Sodium nitrite | Thermo Fisher Scientific | Cat 15633430 |
| MES | Thermo Fisher Scientific | Cat J60763.AP |
| Ethanol | Sigma-Aldrich | Cat 32205-M |
| Triethylammonium acetate | Thermo Fisher Scientific | Cat 90358 |
| Deionized formamide | Roth | Cat P040.1 |
| Antarctic phosphatase | New England Biolabs | Cat M0289S |
| T4 Polynucleotide Kinase | New England Biolabs | Cat M0201S |
| NEBNext Small RNA Library Prep Set for Illumina | New England Biolabs | Cat E7330S |
| NEBNext Multiplex Oligos for Illumina (Index Primer Sets 1 and 3) | New England Biolabs | Cat E7335S and E7710S |
| 8% TBE polyacrylamide gels | Thermo Fisher Scientific | Cat EC6215BOX |
| Lipofectamine 2000 | Thermo Fisher Scientific | 11668019 |
| STM2457 | MedChemExpress | Cat HY-134836 |
| Tris-HCl | Sigma-Aldrich | Cat T1503 |

| Reagent/Resource | Reference or Source | Identifier or Catalog Number |
|---|---|---|
| Sodium chloride | Thermo Fisher Scientific | Cat 15855188 |
| EDTA | Gerbu | Cat 1034 |
| Triton X-100 | Sigma-Aldrich | Cat X100 |
| Complete Protease Inhibitor Cocktail | Roche | Cat 11697498001 |
| Trans-Blot Turbo Transfer Pack | Bio-Rad | Cat 1704158 |
| Milk powder | Gerbu | Cat 1602.0500 |
| Immobilon Western HRP Substrate | Merck | Cat WBKLS0500 |
| Cell Titer-Glo Luminescent Cell Viability Assay kit | Promega | Cat G7571 |
| Caspase-Glo 3/7 Assay kit | Promega | Cat G8091 |
| Methanol | Thermo Fisher Scientific | M-4000-PC17 |
| Crystal violet | Sigma-Aldrich | C3886 |
| **Software** | | |
| Trim Galore | https://github.com/FelixKrueger/TrimGalore | version 0.6.6 |
| GLORI-tools | Liu et al, 2023 | version 1.0 |
| Python | | version 3.10.1 |
| Samtools | Li et al, 2009 | version 1.19 |
| STAR | Dobin et al, 2013 | version 2.7.10a |
| Bowtie | Langmead et al, 2009 | version 1.3.0 |
| bedtools getfasta | Quinlan and Hall, 2010 | version 2.25.0 |
| HOMER | Heinz et al, 2010 | version 4.11 |
| DiffLogo | Nettling et al, 2015 | version 2.20.0 |
| methylSig | Park et al, 2014 | version 1.7.0 |
| ShinyGO | Ge et al, 2020 | version 0.81 |
| HISAT2 | Kim et al, 2019 | version 2.2.1 |
| featureCounts | Liao et al, 2013 | version 2.0.6 |
| DESeq2 | Love et al, 2014 | version 1.42.0 |
| DaPars | Xia et al, 2014 | version 1.0.0 |
| Lifelines | https://joss.theoj.org/papers/10.21105/joss.01317 | version 0.30.0 |
| ImageJ | https://imagej.nih.gov/ij/index.html | |
| ColonyArea | Guzmán et al, 2014 | |
| **Other** | | |
| TissueRuptor II | Qiagen | |
| 2200 TapeStation | Agilent Technologies | |
| Thermocycler T3000 | Biometra | |
| NovaSeq 6000 | Illumina | |

| Reagent/Resource | Reference or Source | Identifier or Catalog Number |
|---|---|---|
| Trans-Blot Turbo Transfer System | Bio-Rad | |
| M6 ECL Chemostar fluorescence imaging system | Intas | |
| GloMax Explorer Multimode Microplate Reader | Promega | |

## Urothelial carcinoma patients and sample acquisition

This study was performed in adherence to the Declaration of Helsinki. All patients provided informed consent to participate in the molecular characterization of their tissue samples. Additionally, approval of the institutional ethics review board (Ethical Committee II, University of Heidelberg, Germany, reference number: 2015-549N-MA) was taken. Patient information is provided in Table EV4.

Nine tumor samples were obtained from transurethral resection of the bladder (TURB) specimens. Nine independent paratumoral control tissues were taken from cystectomy specimens. Samples were diagnosed by an uropathologist, and tumors were characterized according to the TNM classification for bladder cancer by the Union for International Cancer Control (UICC 2017). Bladder tumors with variant histopathological findings other than urothelial carcinoma were excluded. Patient tissue samples were stored at -20 °C in RNAlater solution until further processing.

## Cell culture

Cell lines were authenticated by single nucleotide polymorphism-profiling, tested for mycoplasma and cultured based on ATCC guidelines. HEK293T (RRID: CVCL_0063) and UM-UC-3 (RRID: CVCL_1783) cell lines were cultured in DMEM high glucose medium supplemented with 10% FBS and 1% P/S. RT4 (RRID: CVCL_0036) and T24 (RRID: CVCL_0554) cell lines were cultured in McCoy's 5 A (modified) medium supplemented with 10% FBS and 1% P/S. All cell lines were cultivated as adherent monolayers at 37 °C in a humidified incubator with an atmosphere of 5% $CO_2$.

## Isolation and preparation of RNA samples for GLORI-sequencing

RNA isolation and preparation for GLORI were performed as described (Liu et al, 2023; Shen et al, 2024). Patient tissue samples were homogenized by disruption using the TissueRuptor II (Qiagen). Total RNA from homogenized tissue samples and HEK293T cells was isolated with TRIzol. Small RNA fractions were depleted using the MEGAclear Transcription Clean-Up Kit (Thermo Fisher Scientific). Enrichment of mRNA was performed twice using the Dynabeads mRNA Purification Kit (Thermo Fisher Scientific). mRNA was then fragmented by incubation for 3 min at 94 °C using the NEBNext Magnesium RNA Fragmentation Module (New England Biolabs). The fragmentation step was performed differently from the published version of the protocol which described mRNA fragmentation for 4 min at 94 °C. Fragmented mRNA was then DNase I-treated and

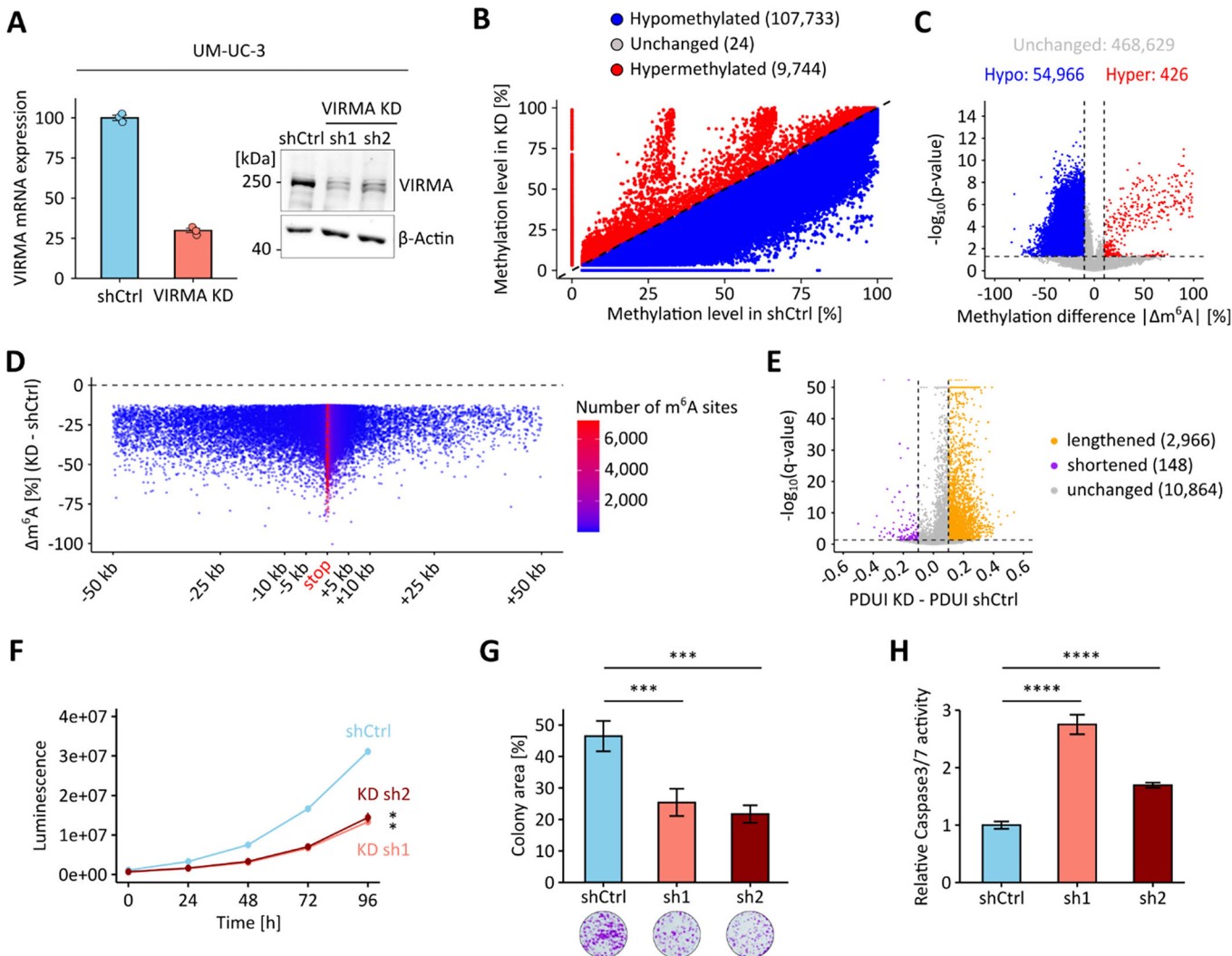

**Figure 6. VIRMA depletion reduces m⁶A methylation and impairs the oncogenic phenotype of UM-UC-3 cells.**

(A) RNA-seq and Western blot analyses of VIRMA expression levels in UM-UC-3 VIRMA KD and shCtrl cells. (B) Scatter plot comparing mean m⁶A levels of methylated DRAC(H) sites between UM-UC-3 VIRMA KD and shCtrl cell lines. (C) Overview of differentially methylated DRAC(H) sites in VIRMA-depleted UM-UC-3 cells, based on |Δmethylation| > 10% and $p < 0.05$ thresholds. Statistical significance was assessed using beta binomial models. Unmethylated DRAC(H) sites are included in this analysis. (D) Δm⁶A levels (UM-UC-3 VIRMA KD - shCtrl) for differentially hypomethylated m⁶A sites were plotted across transcript regions surrounding the stop codon. (E) DaPars-based analysis of APA showing ΔPDUI for UM-UC-3 VIRMA-depleted cells compared to shCtrl cells. These analyses were performed based on $n = 3$ biological replicates. (F) Cell proliferation of UM-UC-3 VIRMA KD and shCtrl cells. sh1*$p = 0.033$, sh2*$p = 0.045$, two-way analysis of variance. (G) Colony formation results from UM-UC-3 VIRMA KD and shCtrl cells. sh1***$p = 0.0007$, sh2***$p = 0.0006$, two-tailed Student's t test. (H) Caspase 3/7 activity measurements in UM-UC-3 VIRMA KD and shCtrl cells. ****$p < 0.0001$, two-tailed Student's t test. Data are represented as mean ± SD; $n = 4$ biological replicates. Source data are available online for this figure.

purified by the RNA Clean & Concentrator-5 kit (Zymo Research). The concentration as well as the average and peak mRNA fragment lengths were determined by TapeStation (Agilent Technologies).

## Protection, deamination, and deprotection of RNA samples for GLORI-sequencing

All steps were performed as described (Liu et al, 2023; Shen et al, 2024). For RNA protection, 100–200 ng of fragmented mRNA supplemented with a synthetic Spike-In RNA oligonucleotide were added into protection buffer (1.32 M Glyoxal solution, 50% DMSO, prepared in water) and incubated for 30 min at 50 °C in a thermal cycler. The sequence of the Spike-In RNA oligonucleotide is listed in

Table EV6. The RNA was further stabilized by adding 10 µL of freshly prepared saturated boric acid at RT, followed by incubation for 30 min at 50 °C in a thermal cycler. Protected RNA was then added into 50 µL of freshly prepared deamination buffer (1.5 M NaNO₂, 80 mM MES (pH 6.0), 1.76 M Glyoxal solution, prepared in water), and incubated for 8 h at 16 °C in a thermal cycler. After deamination, RNA was purified by ethanol precipitation overnight at −80 °C. Pelleted RNA was washed twice using 75% ethanol and air-dried for 5 min at RT. Then, the RNA pellet was resuspended in 50 µL of deprotection buffer (500 mM triethylammonium acetate (pH 8.6), 47.5% deionized formamide, prepared in water) and incubated for 10 min at 95 °C in a thermal cycler. Deprotected RNA was purified by ethanol precipitation for at least 30 min at −80 °C. Pelleted RNA was

washed twice using 75% ethanol and air-dried for 5 min at RT. The RNA pellet was then resuspended in 50 μL of water and further purified using the RNA Clean & Concentrator-5 kit (Zymo Research).

## Preparation of GLORI libraries and sequencing

For sequencing library preparation, a two-step RNA end-repair protocol was used. RNA 3'-end dephosphorylation was performed by Antarctic phosphatase (New England Biolabs) treatment in a total reaction volume of 20 μL. The reaction was incubated for 30 min at 37 °C and inactivated for 2 min at 80 °C. Subsequently, RNA 5'-end phosphorylation was performed by T4 Polynucleotide Kinase (New England Biolabs) treatment in a total reaction volume of 50 μL. The reaction was incubated for 30 min at 37 °C, inactivated for 20 min at 65 °C, and purified using the RNA Clean & Concentrator-5 kit (Zymo Research). GLORI-sequencing libraries were then prepared using the NEBNext Small RNA Library Prep Set for Illumina in combination with the NEBNext Multiplex Oligos for Illumina (Index Primer Sets 1 and 3) (New England Biolabs). Size selection of the libraries was performed via TBE-PAGE using 8% TBE polyacrylamide gels (Thermo Fisher Scientific) selecting sequencing competent molecules in a size range of 160–250 bp. Average peak size and concentration of the libraries were determined by TapeStation (Agilent Technologies). Libraries were sequenced on a NovaSeq 6000 platform (Illumina) applying a 100 bp paired-end sequencing protocol. Sequencing was performed by the Next Generation Sequencing Core Facility of the German Cancer Research Center, Heidelberg. Raw sequencing data were then trimmed using Trim Galore (version 0.6.6, https://github.com/FelixKrueger/TrimGalore) and further processed by the GLORI-tools pipeline as described (Liu et al, 2023; Shen et al, 2024). GLORI-tools is available on GitHub: https://github.com/liucongcas/GLORI-tools. Parameters for the selection of m⁶A sites were set as follows: ≥5 variant nucleotides, ≥15 coverage of A and G bases, ≥0.1 A rate (methylation level >10%). All steps of the GLORI-tools pipeline were executed using the following software: python (version 3.10.1), samtools (version 1.19 (Li et al, 2009)), STAR (version 2.7.10a (Dobin et al, 2013)), bowtie (version 1.3.0 (Langmead et al, 2009)). The human genome (GRCh38) and transcriptome (GCF_000001405.39) reference files were downloaded from UCSC.

## Motif analysis

The coordinates and flanking sequences of GLORI-derived m⁶A sites were extracted using bedtools getfasta (version 2.25.0 (Quinlan and Hall, 2010)) and the human genome reference file (GRCh38). Sequence motifs were then identified using Hypergeometric Optimization of Motif EnRichment (HOMER, version 4.11 (Heinz et al, 2010)). For visualization, the motif representations were converted into position weight matrices using DiffLogo (version 2.20.0 (Nettling et al, 2015)).

## Differential methylation analysis

The R package methylSig (version 1.7.0 (Park et al, 2014)) was used to identify differentially methylated sites comparing control and cancer tissues. For the analysis, an absolute methylation level cutoff of 10% and a *p*-value cutoff of 0.05 were selected. The statistical significance was determined using beta binomial models. Gene ontology analyses were performed using ShinyGO (version 0.81 (Ge et al, 2020)).

## Bulk RNA-sequencing

Total RNA from tissue samples and cell lines was isolated using TRIzol. To remove residual gDNA contaminants, total RNA samples were DNase I-digested and further purified using the RNA Clean & Concentrator-5 kit (Zymo Research). Library preparation and sequencing were performed by the Next Generation Sequencing Core Facility of the German Cancer Research Center, Heidelberg. The samples were sequenced on a NovaSeq 6000 platform (Illumina) applying a 100 bp paired-end sequencing protocol. Reads were trimmed using Trim Galore (version 0.6.6) and mapped to the human reference genome (GRC38) using HISAT2 (version 2.2.1 (Kim et al, 2019)).

## Differential gene expression analysis

Aligned reads from bulk RNA-sequencing experiments were counted by the featureCounts function (2.0.6 (Liao et al, 2013)), normalized, and differential gene expression analysis performed via DESeq2 (version 1.42.0 (Love et al, 2014)). Only transcripts with a read count >10 in every sample were included in downstream analyses.

## Weighted global methylation level analysis

For the determination of the weighted global methylation level in the individual samples, m⁶A methylation data from GLORI-sequencing and transcript expression data from RNA-sequencing were integrated. The methylation level of each transcript was aggregated and then multiplied by the TPM-normalized expression level of the same transcript. These individual, weighted methylation levels were then summed up to compute the weighted global methylation level.

## Analysis of alternative polyadenylation events

For the de novo identification of alternative polyadenylation sites, aligned reads from bulk RNA-sequencing experiments were analyzed using the DaPars algorithm, version 1.0.0 (Xia et al, 2014). For the analyses, a coverage cutoff of 30, an FDR cutoff of 0.05, an absolute PDUI difference cutoff of 0.1, and an absolute fold change cutoff of 0.59 were selected.

## Establishment of T24 and UM-UC-3 METTL3 knockout clones

HEK293T cells were transfected with the lentiviral packaging vectors psPAX (Plasmid #12260, Addgene) and pMD2.G (Plasmid #87360, Addgene) as well as the pLentiCRISPR v2 vector (Plasmid #52961, Addgene) using Lipofectamine 2000 (Thermo Fisher Scientific). Anti-METTL3 and scramble sgRNA sequences are listed in Table EV6. Transfected HEK293T cells were incubated for 48 h. T24 and UM-UC-3 cells were transduced for 48 h and further cultivated for the establishment of clonal populations.

## STM2457 treatment of urothelial carcinoma cells

T24 and UM-UC-3 cells were cultured until reaching a confluency of approximately 80%. The cells were then treated for 48 h with DMSO or 50 μM of the METTL3 inhibitor STM2457 (MedChemExpress).

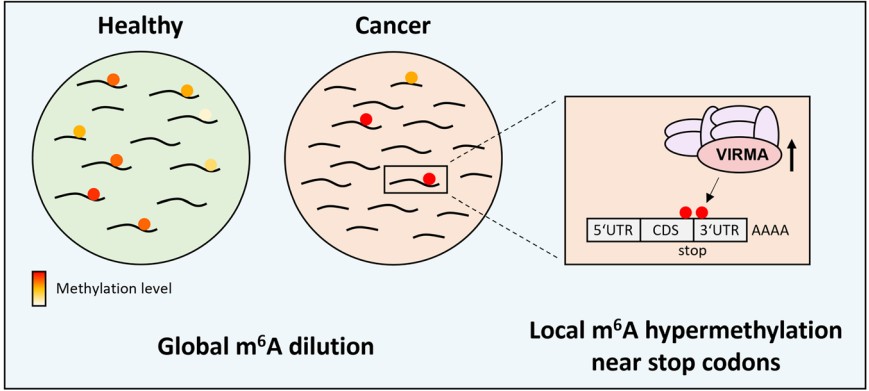

**Figure 7. The m⁶A epitranscriptomic landscape of bladder cancer is characterized by global methylation dilution and focal 3′-UTR hypermethylation.**

Globally, the upregulation of unmethylated transcripts and the downregulation of highly methylated transcripts resulted in the dilution of m⁶A methylation levels in bladder cancer. Our findings further suggest that the upregulation of VIRMA causes the local hypermethylation of m⁶A modified target transcripts in regions close to the stop codon.

## Databank VIRMA expression analysis

VIRMA mRNA expression data were downloaded from the UCSC database for the combined The Cancer Genome Atlas bladder urothelial carcinoma (TCGA-BLCA) cohort and the Genotype-Tissue Expression (GTEx) project (https://xenabrowser.net/). Expression data were DESeq2-normalized and $\log_2$-transformed ($\log_2$(value + 1)). Mann–Whitney U test was assessed to test for statistical significance between control and UCB tissues.

## Association analysis of VIRMA genomic alterations and expression

Genetic alteration data and the corresponding mRNA expression data for VIRMA were downloaded from the cBioPortal database (https://www.cbioportal.org/) for TCGA-BLCA cohort. VIRMA mRNA expression values were preprocessed from RNA-sequencing by Expectation-Maximization (RSEM) data generated by the TCGA RNASeqV2 pipeline (Illumina HiSeq), which were batch-normalized and $\log_2$-transformed ($\log_2$(value + 1)). Different copy-number alterations (shallow deletion, diploid, gain, amplification) were identified based on the Genomic Identification of Significant Targets in Cancer (GISTIC) method. Pearson correlation analysis was performed to assess the association between $\log_2$ copy-number values and VIRMA mRNA expression levels.

## Survival analysis

Kaplan–Meier overall survival analysis was performed using the lifelines python package. Patients were stratified into VIRMA-high and VIRMA-low mRNA expression groups using maximally selected log-rank statistics.

## Establishment of VIRMA knockdown cell lines

Lentiviral shRNA constructs for VIRMA KD were purchased from Horizon Discovery (clone_96736 (RHS4430-200156255) and clone_96733 (RHS4430-200172168). The sequences of the anti-

VIRMA shRNAs are listed in Table EV6. Sequence information for the non-targeting shRNA control was not provided by the manufacturer. Transfection and transduction were performed as described for the establishment of METTL3 KO cells.

## Western blot

Cells were lysed in 20 mM Tris-HCl (pH 7.5), 150 mM NaCl, 1 mM EDTA and 1% Triton X-100 supplemented with the Complete Protease Inhibitor Cocktail (Roche, Basel, Switzerland). Proteins were then separated by SDS-PAGE and transferred to nitrocellulose membranes using a Trans-Blot Turbo Transfer System (Bio-Rad). Membranes were blocked in 0.1% PBST containing 5% milk powder for 1 h at room temperature. Primary antibody incubation was performed overnight at 4 °C using the β-Actin monoclonal antibody (Sigma, A5316) and the VIRMA polyclonal antibody (Proteintech, 25712-1-AP). Secondary antibody incubation occurred for 1 h at room temperature. Finally, membranes were imaged using the Immobilon Western HRP Substrate (Merck), and signals detected using an M6 ECL Chemostar fluorescence imaging system (Intas).

## Cancer cell phenotypic assays

For cell proliferation assays, VIRMA KD and shCtrl cells were seeded in 96-well plates with the following densities: UM-UC-3 cells: 1000 cells/well. RT4 cells: 1500 cells/well. Cell proliferation was quantified by Cell Titer-Glo (Promega) measurements for 5 consecutive days in time intervals of 24 h. Two-way analysis of variance was used to test for statistical significance between VIRMA KD and shCtrl cells. For colony formation assays, VIRMA KD and shCtrl cells were seeded in 6-well plates with the following densities: UM-UC-3 cells: 500 cells/well. RT4 cells: 1000 cells/well. Then, cells were incubated for 1.5 weeks and fixed with ice-cold methanol for 10 min. Staining was conducted using a 0.5% crystal violet solution for 10 min at room temperature. Colonies were counted using the ImageJ plugin "ColonyArea" (Guzmán et al, 2014). Two-tailed Student's t tests were used to test for statistical

significance between VIRMA KD and shCtrl cells. For apoptosis assays, VIRMA KD and shCtrl cells were seeded in 96-well plates with 10,000 cells/well. Caspase activity was quantified by the Caspase-Glo 3/7 Assay kit (Promega) 24 h after seeding. Also, a Cell Titer-Glo (Promega) measurement was conducted to determine the number of cells for normalization. Two-tailed Student's t tests were used to test for statistical significance between VIRMA KD and shCtrl cells.

## Data availability

Data generated in this study have been deposited in the GEO database under the accession numbers GSE281749 and GSE281750.

The source data of this paper are collected in the following database record: biostudies:S-SCDT-10_1038-S44319-026-00739-y.

## Peer review information

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

## Acknowledgements

We thank the Genomics and Proteomics Core Facility of the German Cancer Research Center, particularly Franziska Petermann and Panagiotis Provataris for their support. We also thank Sandra Blanco for critically reading the manuscript. This work was supported by a grant from Deutsche Forschungsgemeinschaft (DFG, German Research Foundation) to FL (project number 439669440 TRR319 RMaP TP A01). Additional funding was provided by the DKFZ-Hector Seed Funding Program (project UCGLORI).

## Author contributions

**Jonas Koch**: Data curation; Software; Formal analysis; Validation; Investigation; Visualization; Methodology; Writing—original draft. **Jinyun Xu**: Formal analysis; Investigation; Visualization. **Felix Bormann**: Software; Formal analysis; Validation; Investigation; Visualization. **Vitor Coutinho Carneiro**: Supervision. **Manuel Neuberger**: Resources; Funding acquisition; Writing—review and editing. **Katja Nitschke**: Resources. **Malin Nientiedt**: Conceptualization; Resources; Funding acquisition; Writing—review and editing. **Philipp Erben**: Resources; Writing—review and editing. **Maurice Stephan Michel**: Resources. **Manuel Rodriguez-Paredes**: Supervision; Funding acquisition; Writing—review and editing. **Frank Lyko**: Conceptualization; Resources; Data curation; Supervision; Funding acquisition; Writing—original draft; Project administration.

Source data underlying figure panels in this paper may have individual authorship assigned. Where available, figure panel/source data authorship is listed in the following database record: biostudies:S-SCDT-10_1038-S44319-026-00739-y.

## Funding

## Disclosure and competing interests statement

The authors declare no competing interests.

# Expanded View Figures

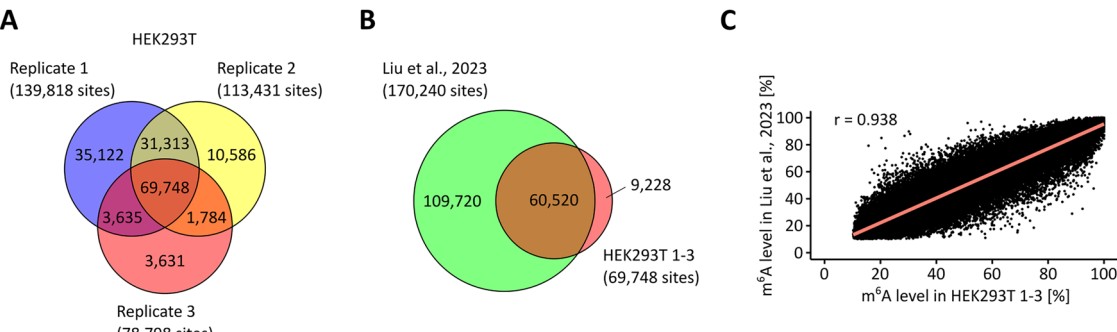

**Figure EV1. GLORI allows for the reproducible detection and quantification of m⁶A sites in HEK293T cells.**

(A) In the intersection of the three HEK293T cell replicates, roughly 70,000 m⁶A sites were detected by GLORI. (B) The majority (87%) of those 70,000 m⁶A sites were also reported by the original study. (C) The mean m⁶A methylation levels from the 60,000 m⁶A sites detected in both studies were comparable.

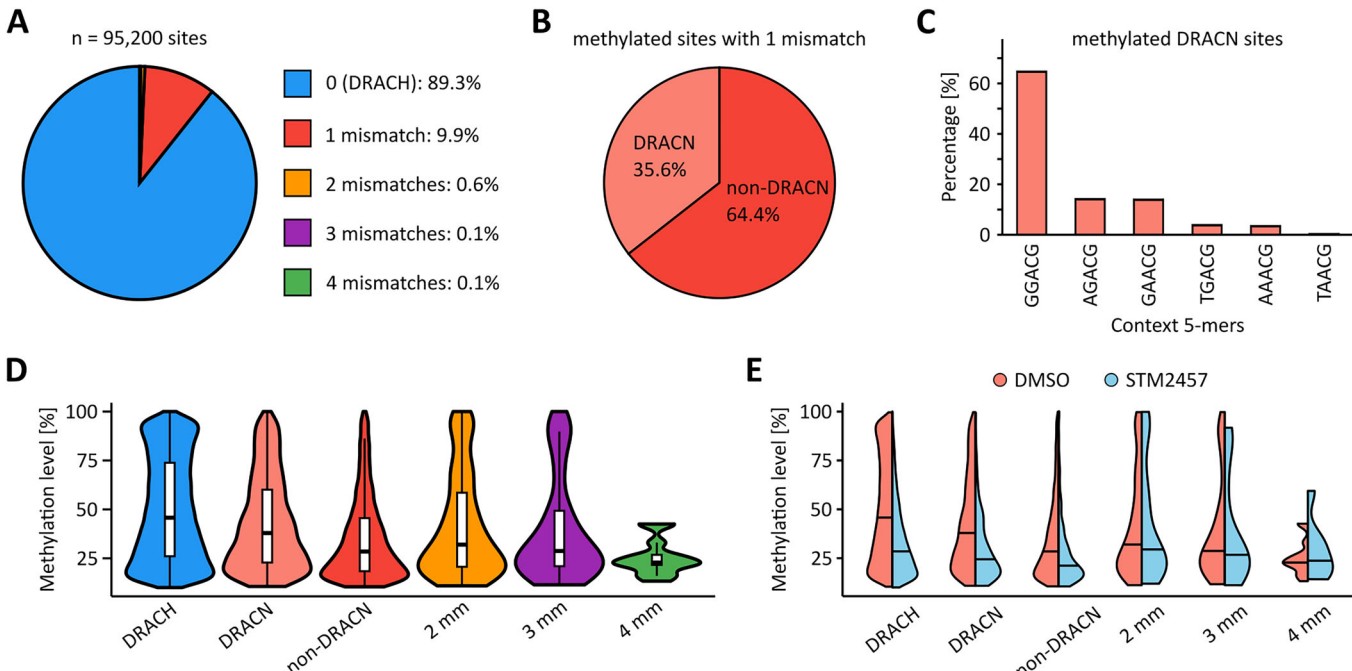

**Figure EV2.   m⁶A sites detected in DRAC(H) motifs are the most robust.**

(**A**) m⁶A sites detected in different sequence motifs. Numbers describe mismatches compared to the canonical DRAC(H) motif. (**B**) Distribution of m⁶A sites detected in 5-mer motifs with one mismatch compared to DRAC(H). In DRACN motifs, the fifth base of the 5-mer is variable, while the first four bases are DRAC(H)-conform. In non-DRACN motifs, the mismatch occurs in one of the first four bases. (**C**) Frequency of DRACN motifs detected in T24 cells. (**D**) Methylation level distribution of m⁶A sites detected in the different motif categories. $n = 95,200$ m⁶A sites shared among the three T24 replicates were used for sequence motif analyses. Boxplots were generated in R using ggplot2. The centre line indicates the median (50th percentile). The box bounds represent the first and third quartiles (25th and 75th percentiles), with box height equal to the IQR. Whiskers extend to the most extreme values within 1.5 * IQR from the quartiles. (**E**) Methylation level distribution of m⁶A sites detected in the different motif categories in T24 cells treated with DMSO or the METTL3 inhibitor STM2457. mm = mismatch compared to DRAC(H). $n = 95,200$ m⁶A sites shared among the three T24 DMSO replicates and $n = 17,189$ m⁶A sites shared among the three T24 STM2457 replicates were used for this sequence motif analysis.

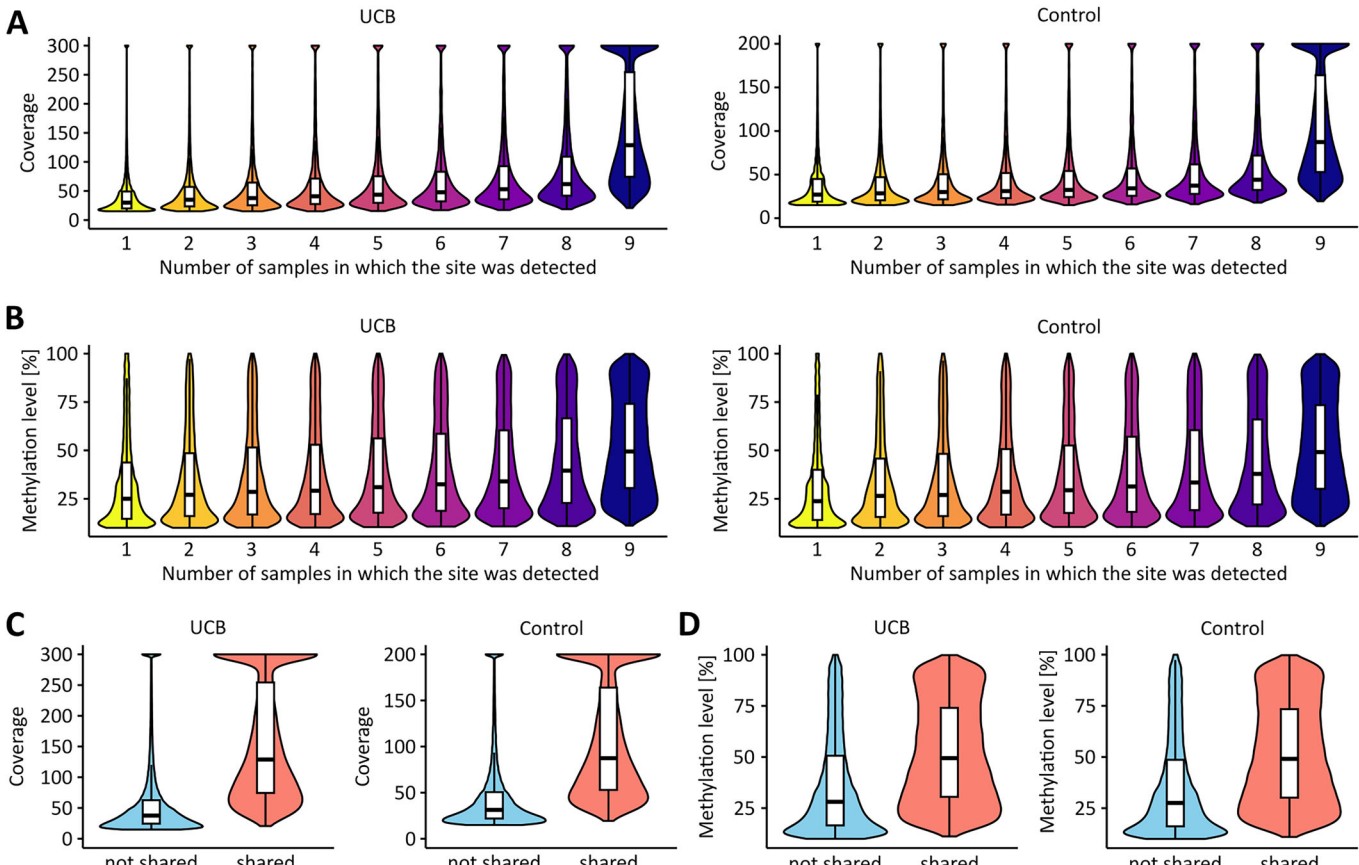

**Figure EV3. Shared m⁶A sites are characterized by high coverage and high methylation levels.**

(A) m⁶A sites were grouped based on the number of samples in which they were detected (from 1 to 9), and the distributions of sequencing coverage are shown for each detection category. (B) m⁶A sites were grouped based on the number of samples in which they were detected (from 1 to 9), and the distributions of methylation levels are shown for each detection category. (C) m⁶A sites detected in all nine samples (shared) were compared against those detected in only 1–8 samples (not shared), and the distributions of sequencing coverage are shown. (D) m⁶A sites detected in all nine samples (shared) were compared against those detected in only 1–8 samples (not shared) and the distributions of methylation levels are shown. $n = 242{,}371$ sites detected in nine UCB tissue samples and $n = 191{,}722$ sites detected in nine control tissue samples were used for these analyses. Boxplots were generated in R using ggplot2. The centre line indicates the median (50th percentile). The box bounds represent the first and third quartiles (25th and 75th percentiles), with box height equal to the IQR. Whiskers extend to the most extreme values within 1.5 * IQR from the quartiles.

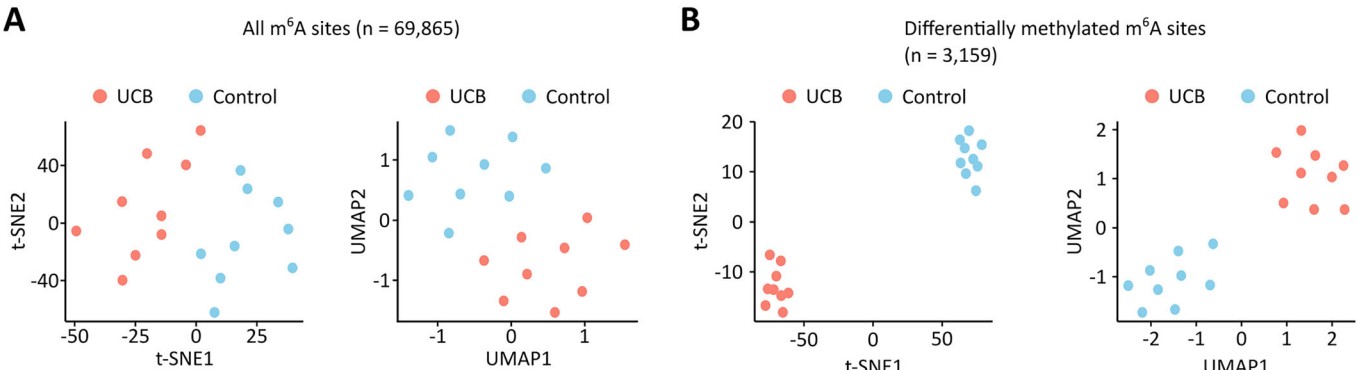

**Figure EV4. tSNE and UMAP analyses separating control and UCB tissue samples based on their m⁶A signatures.**

(A) Dimensionality reduction analyses considering all m⁶A sites. (B) Dimensionality reduction analyses considering differentially methylated m⁶A sites. These analyses were performed based on $n = 9$ biological replicates.

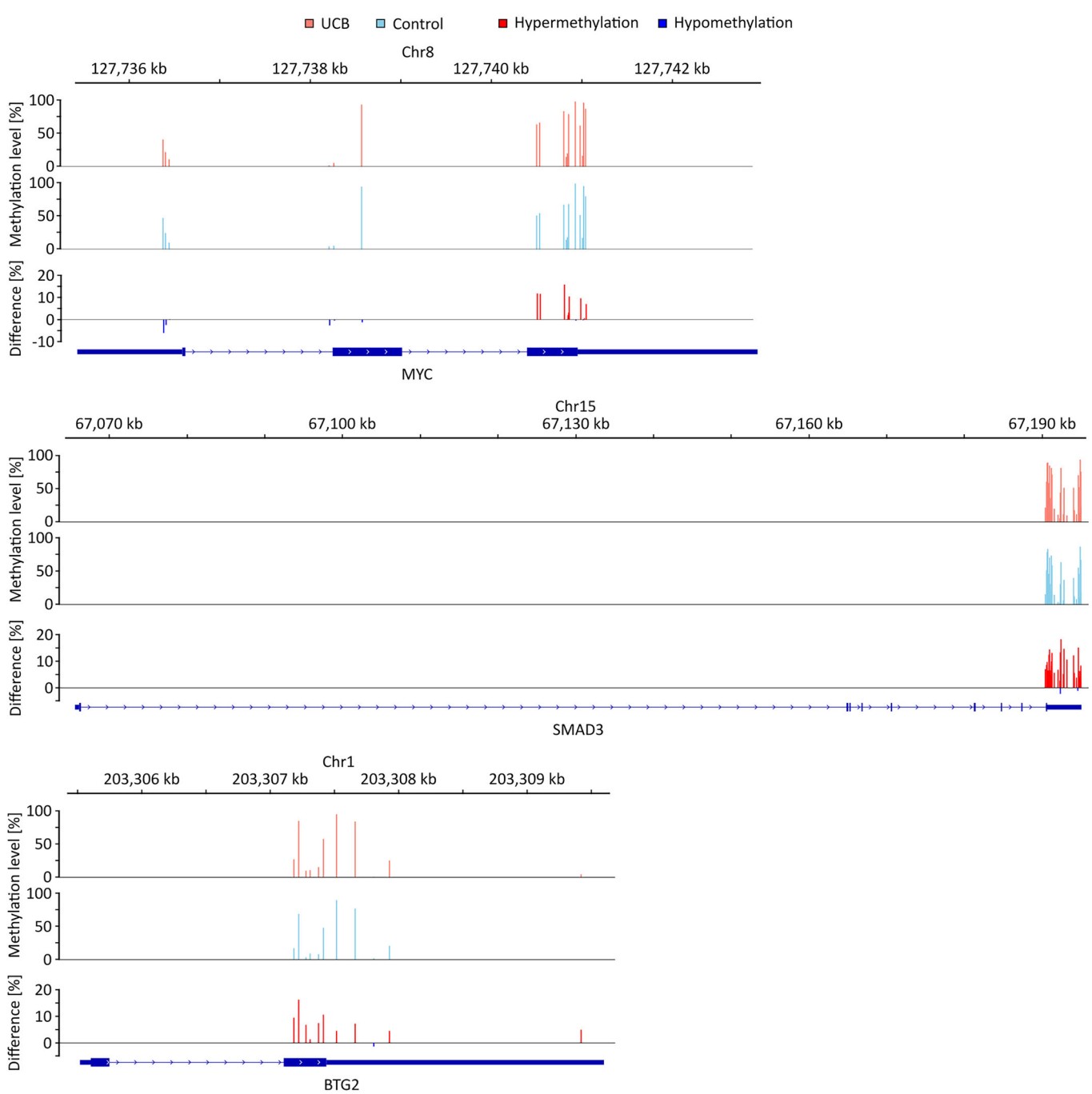

**Figure EV5. Methylation level changes in selected transcripts.**

IGV browser tracks of MYC, SMAD3, and BTG2 transcripts showing altered methylation levels in m⁶A sites comparing control and UCB tissues. These analyses were performed based on $n = 9$ biological replicates.

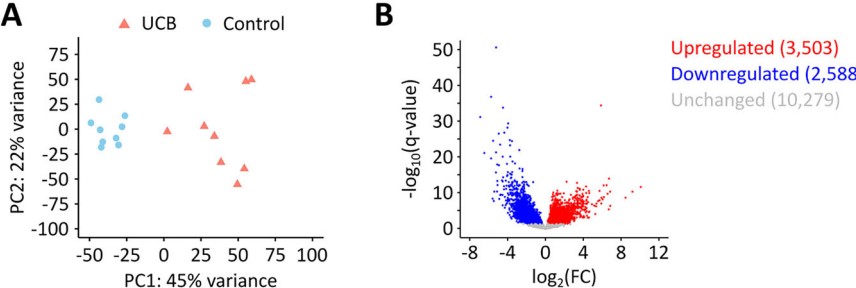

**Figure EV6.   RNA sequencing analysis of clinical samples.**

(**A**) PCA demonstrating that patient samples can be separated based on their gene expression profile. (**B**) Differential gene expression analysis showing global deregulation of genes in UCB. q-value < 0.05. These analyses were performed based on $n = 9$ biological replicates.

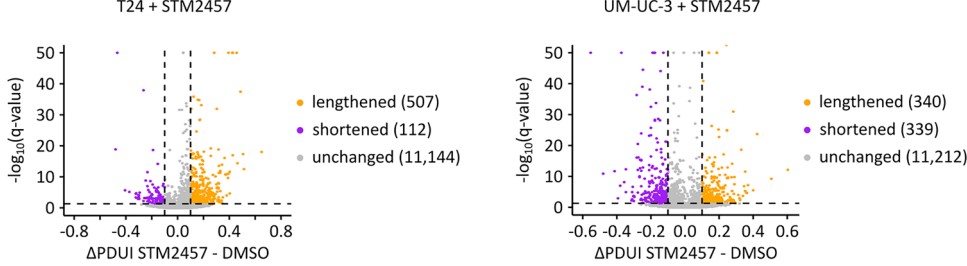

**Figure EV7. Detection of alternative polyadenylation events in two UCB cell lines treated with the METTL3 inhibitor STM2457.**

In T24 cells, most transcripts were found to be lengthened, while no clear tendency was observed in UM-UC-3 cells. These analyses were performed based on $n = 3$ biological replicates.

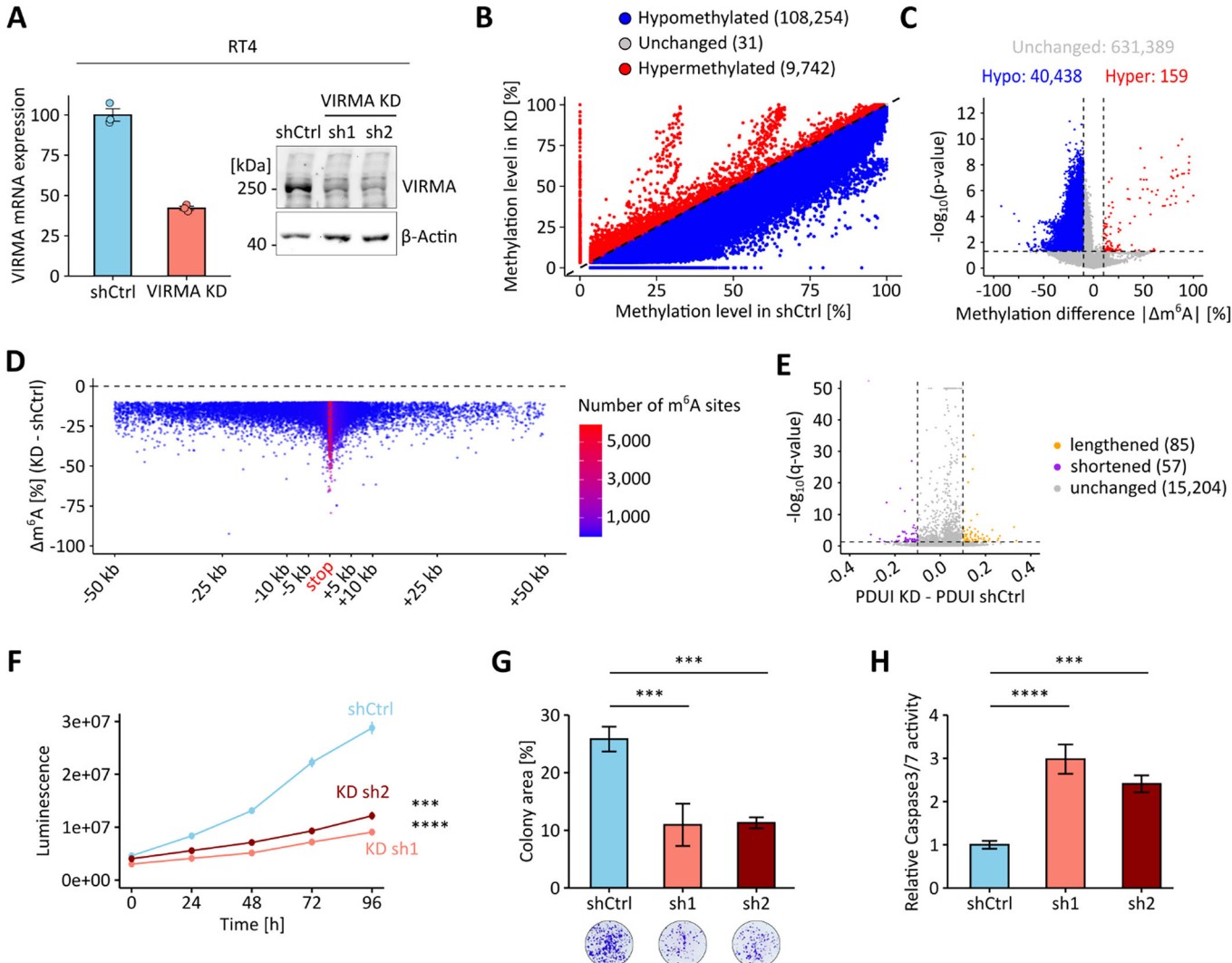

**Figure EV8. VIRMA depletion reduces m⁶A methylation and impairs the oncogenic phenotype of RT4 cells.**

(A) RNA-seq and Western blot analyses of VIRMA expression levels in RT4 VIRMA KD and shCtrl cells. (B) Scatter plot comparing mean m⁶A levels of methylated DRAC(H) sites between RT4 VIRMA KD and shCtrl cell lines. (C) Overview of differentially methylated DRAC(H) sites in VIRMA-depleted RT4 cells, based on |Δmethylation| > 10% and $p < 0.05$ thresholds. Statistical significance was assessed using beta binomial models. Unmethylated DRAC(H) sites are included in this analysis. (D) Δm⁶A levels (RT4 VIRMA KD - shCtrl) for differentially hypomethylated m⁶A sites were plotted across transcript regions surrounding the stop codon. (E) DaPars-based analysis of APA showing ΔPDUI for RT4 VIRMA-depleted cells compared to shCtrl cells. These analyses were performed based on $n = 3$ biological replicates. (F) Cell proliferation of RT4 VIRMA KD and shCtrl cells. sh1****$p < 0.0001$, sh2***$p = 0.0008$, two-way analysis of variance. (G) Colony formation results from RT4 VIRMA KD and shCtrl cells. ***$p = 0.0007$, two-tailed Student's t test. (H) Caspase 3/7 activity measurements in RT4 VIRMA KD and shCtrl cells. sh1****$p < 0.0001$, sh2***$p = 0.0001$, two-tailed Student's t test. Data are represented as mean ± SD; $n = 4$ biological replicates. Source data are available online for this figure.

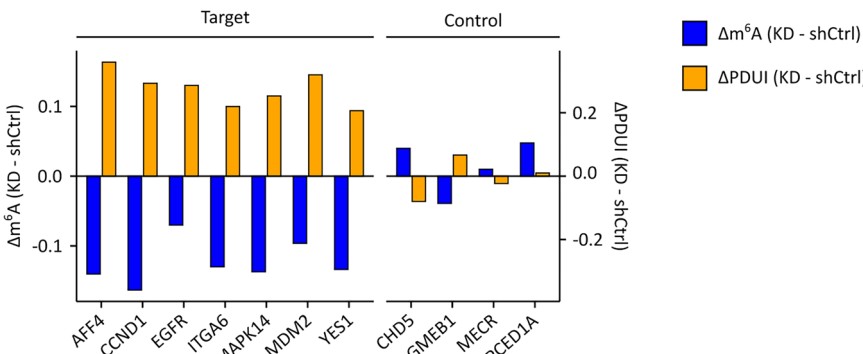

**Figure EV9. VIRMA KD-associated changes in m⁶A methylation and alternative polyadenylation at the transcript level.**

Bar plots show changes in m⁶A methylation ($\Delta$m⁶A; left y-axis) and polyadenylation site usage ($\Delta$PDUI; right y-axis) upon VIRMA KD in UM-UC-3 cells. $\Delta$m⁶A was calculated as the mean m⁶A methylation level across all identified m⁶A sites within the terminal exon and 3'-UTR of each transcript. Target genes show reduced m⁶A methylation accompanied by increased $\Delta$PDUI values, indicative of transcript lengthening. $\Delta$m⁶A and $\Delta$PDUI values are shown relative to shCtrl conditions, $n = 3$ biological replicates.

