## [Peer Review File · EMBO Reports]

The bladder cancer m6A landscape is defined by global methylation dilution and focal 3'-UTR hypermethylation

Jonas Koch, Jinyun Xu, Felix Bormann, Vitor Coutinho Carneiro, Manuel Neuberger, Katja Nitschke, Malin Nientiedt, Philipp Erben, Maurice Michel, Manuel Rodríguez-Paredes, and Frank Lyko

Corresponding author(s): Frank Lyko (f.lyko@dkfz-heidelberg.de)

Review Timeline:

Submission Date:	9th Jul 25
Editorial Decision:	24th Sep 25
Appeal Received:	30th Sep 25
Editorial Decision:	1st Oct 25
Revision Received:	5th Dec 25
Editorial Decision:	20th Jan 26
Revision Received:	2nd Feb 26
Accepted:	24th Feb 26

Editor: Esther Schnapp

Transaction Report:

23rd Sep 2025

Dear Dr. Lyko,

Thank you for your patience while your manuscript was peer-reviewed at EMBO reports. We have now received the full set of referee reports and I am sorry to say that the evaluation of your study is not a positive one.

As you will see, while referee 2 is overall more positive and referee 1 acknowledges that your dataset is of high technical quality and a useful resource for the community, all referees together list a number of concerns that preclude the publication of your work by EMBO reports as it stands now. We are not confident that the different issues raised can be addressed within our standard revision time of 3-4 months.

Referee 1 remarks that no firm conclusions can be drawn regarding (1) the basis for the discrepancy between mass-spec and genomics (2) a VIRMA-mediated regulon of hypermethylated stop codons leading to alternative polyadenylation and (3) a link to UCB cancer progression. Both referees 1 and 3 point out that the reported associations are weak, and both referees rate the novelty and general interest of the study "low" in our ms summary table that is directly sent to the editor.

Referee 2 raises mainly technical issues and agrees with referee 1 that the data are not sufficiently strong to support the claim that m6A has widespread effects on APA.

Referee 3 finally agrees with referee 2 that it is not clear whether the small differences observed are biologically meaningful.

Given these comments from 3 experts in the field, I am sorry to say that we have decided that we cannot offer to publish your ms.

While we cannot pursue this manuscript further, we encourage you to transfer your study to our not-for-profit open-access sister journal, Life Science Alliance (LSA). We shared your manuscript and the accompanying reviews with LSA Executive Editor, Tim Fessenden, who is interested in these findings. He is pleased to offer consideration of this manuscript at LSA pending the following revisions: - Temper conclusions on dilution of m6A modifications given the significant methodological concerns of Reviewers 1 and 3, in particular the absence of mass spec measurements of m6A levels. We suggest highlighting the overall global and modest changes in m6A (as remarked by Reviewers 2 and 3) as an important result in itself. - Temper conclusions on the association between polyadenylation and m6A modification, which remains speculative per Reviewers 1 and 2. - Clarify methodology as requested by Reviewer 2: how m6A sites were called (point 1) and on the threshold of 10% change in stoichiometry (point 6) We understand that such a revision might need to be re-reviewed, in which case Dr. Fessenden will walk the Reviewers through our transfer process. We encourage you to use the link below to transfer your manuscript to LSA. You do not need to revise the manuscript before transferring it to LSA. Once you transfer, Dr. Fessenden will email you an invitation to revise and resubmit, listing the same revision requests as mentioned above. Please feel free to reach out at t.fessenden@life-science-alliance.org if you have any questions about the LSA journal, the transfer process, or the revisions requested.

I am sorry that I cannot be more positive for EMBO reports this time and hope that you find the transfer suggestion a good option.

Kind regards,
Esther

Referee #1:

In this manuscript, Koch et al. present a large, single-nucleotide-resolution m6A dataset obtained using GLORI-seq from paired urothelial carcinoma and matched normal bladder tissues. The authors carefully produce and characterize a dataset of high technical quality. The characterization of this dataset identifies features of m6A that have been previously characterized (selection for certain motifs, enrichment near stops) and generally does not identify prominent differences between cancer and normal tissues; Nonetheless, the dataset is a useful resource for the community. The authors then focus on three main aspects: (1) The authors try to explain a discrepancy between mass-spec data, pointing at a global reduction of m6A levels in cancer, and the genomic dataset, that generally fails to find consistent differences. The authors suggest that the changes are a consequence of differences in the composition of the transcriptome ('dilution'), wherein less methylated and highly abundant genes are expressed higher in cancer, thereby leading to overall decreased m6A levels. (2) The authors identify sites enriched around the stop codon that are hypermethylated in cancer tissue. The authors link this hypermethylation around stop codons to elevated

VIRMA expression in cancer tissue. (3) the authors suggest that such methylation may regulate polyadenylation and may be associated with cancer progression.

We find this manuscript of moderate interest. On the one hand, the dataset was carefully assembled and analyzed. On the other hand, the degree of novelty in this manuscript is relatively limited, and the drawn conclusions are often based on relatively weak, associative analyses. Based on the analyses that are performed, it is in our opinion difficult to draw firm conclusions regarding (1) the basis for the discrepancy between mass-spec and genomics, (2) a VIRMA-mediated regulon of hypermethylated stop codons leading to alternative polyadenylation, or (3) a link to UCB cancer progression.

Specific comments:

'm6A dilution model' - in previous work the authors found substantial decreases in m6A levels between tumor and paratumor tissues, with m6A levels being reduced on average more than 50%. Here the authors compare tissues between cancer patients and controls, and find no differences between the two based on GLORI-based readouts. The authors view this as a contradiction, that they seek to resolve by weighting m6A stoichiometries by the expression levels of genes. There are multiple issues with this line of thought, and with the analyses. First, it is not even clear to use that there is any discrepancy between the results. A comparison of tissues to paratissues (previous paper) is very different from a comparison of cancerous tissue from one patient to healthy tissue from another. Perhaps there is simply no difference between the latter (that could be reproducibly detected based on both techniques), but there are considerable differences between the former. Second, given the centrality of this point, a crucial element this manuscript lacks is mass-spec measurements of m6A levels, to match the GLORI data, to allow evaluating whether truly there is a discrepancy. Third, the current conclusion is based on looking at the average weighted methylation levels across groups which qualitatively (but not quantitatively) shows a similar trend as observed in the past study (for other tissues). A much more convincing analysis, demonstrating that discrepancy between mass-spec studies and genomic ones can be bypassed using the approach suggested by the authors, would be to measure m6A via mass-spec for each of the samples (point 2 above) and then to correlate this against the GLORI based index. This is crucial, because mass-spec and genomic data can fail to correlate also for an abundance of other reasons, many of which completely technical (e.g. failure to fully remove rRNA contaminants).

3' UTR hypermethylation associated with APA: The authors observe hypermethylated sites in cancer, and associate them with APA. We find these associations very weak. The authors demonstrate a (mild) enrichment towards decreased 3' UTR length in cancer, and they demonstrate that loss of METTL3 leads to a similar phenomenon. The authors do not establish that these changes in 3' UTR occur within the same genes. Moreover, all the analyses are presented based on average m6A levels, despite the fact that substantial differences in m6A levels are present across tumor and control samples (Fig. 4B). A much more convincing analysis would be to rank samples based on m6A levels at individual sites, and establish that APA levels correlate with this. We should further highlight that in making their point about APA, the authors rely on a previous study (Yue, 2018) which itself makes only cautious claims about the relationship between m6A methylation and polyadenylation. In our reading of the literature the link between m6A and polyadenylation is tenuous at best, and it would be upon the authors to firmly establish it. APA associated with VIRMA: Unfortunately we found these analyses unconvincing as well. First, it is confusing aspect is that the authors attribute the effect to VIRMA (rather than to the entirety of the methylation machinery), and yet in their experiments choose to perturb METTL3 (and even report an expected effect). Second, results are again based on average levels across tumors/controls, whereas they would be much more convincing if the full power of 18 different samples could be used to establish an actual correlation between VIRMA levels and APA. The Kaplan-Meier plots appear to arbitrarily assign 345 samples to one group and 65 to another, and do not give the sense of being very robust.

Additional points:

Definition of Hyper- and Hypomethylated Transcripts: Clarification is required on how transcripts were classified as hyper- or hypomethylated. The manuscript currently lacks details regarding thresholds used and whether the classification is based on single or multiple sites per transcript. It should also be explicitly stated whether these sites behave consistently across entire transcripts or individually.

Panel 5E: what would the corresponding panel look like for transcripts with positive delta-PUI? Is this truly an enrichment? Or just reflection of how m6A is distributed across genes?

Overall, the language in manuscript is strong, whereas links are weak and associative. Language must be moderated.

Differential Gene Expression: The manuscript should clarify how log-fold changes (logFC) of values as extreme as 10 or 20 were derived, as these appear unusually high.

Referee #2:

This is an excellent study on m6A and cancer, using innovative, highly quantitative m6A mapping methods. As a result of using higher quality, mapping methods, many new insights are obtained. In contrast to previous studies, the authors find global and general effects of cancer on m6A rather than highly specific effects. They localized changes to m6A distribution to specific regions of transcript body rather than globally throughout the transcript. They find a general similarity between the normal and the cancer tissue. The study points out that some of the previous work, which is quite a sizable body of literature, may need reevaluation when re-examined using these quantitative methods. I am generally supportive of the study, but at the same time, I think the study needs to talk more about some of the statistical methods that are used to ensure that the differential sites are in fact truly differential and not due to statistical noise. I don't think further wet experimentation is needed, but Some deeper consideration to the computational aspect should be discussed.

1. I have a suggestion about how the discussion of the m6A sites detected using GLORI is described. The author say they have 88,000 sites. But the number of sites that you get in this method is very much dependent upon the threshold that you place on the number for independent reads required to call a site and the number of nonconversions needed. Since there are a substantial number of non-conversions that just occur at baseline, sometimes these putative sites might just be part of the background noise. Multiple independent non-conversions is the most convincing when calling a site. So, the author should explain what their criteria are. Additionally, of those 88,000, they mentioned that they occasionally get noncanonical sites. They attributed these to background. But they only show one of these sites and that site looks very similar to DRACH, which makes me think that these are real m6A events. How many sites were completely unrelated to DRACH? A problem with the study is that there is no great control possible for the samples. There's no knockout cell line. But there needs to be better ways to understand what the background rates are. For instance, what is the non-conversion rate at sites that are unrelated to DRACH? If the non-conversion rate is one percent at these sites, it informs us about what type of false positive rate you might eventually get with that 88,000 called sites. There may be better ways to establish background. The inhibitor is a great and useful control, but it is problematic because the drug does not completely inhibit the enzyme and you don't know if the remaining m6A sites are real or not. This is why other methods need to be considered for establishing the background rate.
2. The authors mentioned improvements to GLORI. Are these similar to the ones used for GLORI2? It is worth mentioning this newer method and how it compares to the authors changes. Additionally, as part of the quality control discussion, it's really important to discuss. The minimum read length threshold that was placed for reads that were analyzed, but also a histogram should be presented for the distribution of read length sizes that were obtained and the threshold for use in the study. This can help us understand how many fragments were too small to be used and how much fragmentation was obtained in general. The authors might want to compare that to the size distribution from the original study. This helps to demonstrate that the methods that were used improved the overall protocol.
3. The introduction talks about 18 cancer samples. But it was unclear to me from the phrasing if it was 18 samples total, some of which were cancer some of which were control or if it was 18 paired cancer and normal tissues.
4. The study impact could be enhanced if the abstract and possibly even the title brings up the exciting conceptual questions that were tested in the study especially the idea that the authors bring up about focal changes versus global changes. The authors are quite correct that so many papers argue for focal changes, but this study really shows broad changes that are not highly selective for specific transcripts. This should really be emphasized because it is a major conceptual advance. Currently the way the title is written, it sounds like the study may have be of interest only to people who care about bladder cancer.
5. The authors discuss about 42,000 shared sites. But that implies that some sites were not shared. However, it is my suspicion that the sites that were not shared simply did not have enough read coverage in one sample versus the other. However, one could read this text to think that there are sites that are unique to one sample or to the control tissue that are simply not found in the bladder cancer despite sufficient coverage. An alternative possibility is that there may be a single non-conversion event in one sample, but not in another and that non-conversion event is not biologically meaningful. However, this would come across as a site ghat is unique in one sample versus other. I think greater clarity and perhaps an assessment of the sites that are shared after considering sufficient depth and creating a minimal threshold of the number of non-conversions that are required for calling an m6A that would be very helpful. I think based on this, I think the authors will conclude that all sites are shared between the samples. If so, this should be stated.
6. The criteria that the authors bring up is a 10% change in stoichiometry. But this seems extremely small. The author should discuss if they think that this type of change is likely to be biologically meaningful. I doubt it, but the author should mention and discuss this. To me, it would be worthwhile to indicate how many sites differ by 20% or 30% or 40%. It's important to remove Sites that have insufficient coverage. For instance, if there are two different samples and they each have a coverage of 10 reads, and the ground truth is 10% stoichiometry, simple statistical variation could allow one sample to have one out of 10 non-conversions and the other with two out of 10 non-conversions but this could both refer to 10%. When the authors talk about the number of sites that vary by 10%, is there sufficient read depth to know that this isn't simply statistical noise? I would suggest that they use or at least discuss the rigid criteria needed to know that these 10% stoichiometry changes are in fact real and not statistical noise. The text should talk much more about the read threshold numbers that are needed in order to be confident that a change in the m6A level is likely to be real rather than potentially due to random noise. For different sites with different levels of coverage, the m6A level would have to change to different degrees in order for that change to be meaningful. For instance of the site has 100 reads of coverage, a 10% change would be quite impressive. But if there were only 10 reads, that 10% might just be noise.
7. Related to the point above, in the examples that are shown, it seems that all the sites seem to be elevated in the m6A level. Is this the trend that is seen? That an mRNA tends to show that all the sites are elevated or all the sites are decreased? Or do some sites go up in some sites go down?
8. The dilution concept was quite interesting and novel, and the results about this really caste doubt on whether researchers really should draw deep interpretations from mass spectrometry experiments. So this was a very interesting part of the study. But when I read the abstract, it wasn't clear to me that the dilution wasn't caused by transcripts that are normally methylated, but now are not methylated. I think it's not clear. only when you read the text, you realize that these are from transcripts that are normally not methylated and the overall expression changes.
9. There is some discussion that previous studies used antibody based methods to do mapping. It should be pointed out that this method was not designed to be quantitative. It was designed to be qualitative. However, people used it in a quantitative manner without doing the proper Spike in standards are not using correct methods like m6A-seq2 which is more quantitative.
10. The m6A and APA discussion is somewhat complicated. Keep in mind that Darnell did a study which showed that there were

some small but measurable effect of m6A and APA. But these were very rare transcripts. A recent study by Ries et al. NSMB also found rare examples of m6A depletion leading to shorter 3'UTR. The Moline study is somewhat difficult to interpret because the method also looks at RNA's that contain m6Am, not just m6A so it's difficult to draw complete conclusion from that study. I would recommend caution when making the claim that m6A has clear widespread effects on APA.

Referee #3:

Koch et al mapped m6A signal in some UCB samples (urothelial carcinoma of the bladder) by GLORI-sequencing, a single-nucleotide resolution m6A-mapping method previously established by other labs. The authors claimed that 1) the global loss of m6A signal in UCB is due to increased expression of unmethylated transcripts and decreased expression of highly methylated transcripts; 2) local 3UTR hypermethylation was associated with the overexpression of VIRMA; Both claims are based on superficial correlations and there is neither mechanistic nor clinical insights in the work, thus it is not suitable for the publication at EMBO. Some major issues include the following:

- 1) The entire analysis of this work is based on very small differences (i.e. delta m6A change >10% and $p < 0.05$ line 156), and there is no convincing data to support that such small change of m6A signal has a biological impact.
- 2) The claim of "the global loss of m6A signal in UCB is due to increased expression of unmethylated transcripts and decreased expression of highly methylated transcripts" itself is a superficial and hypothetical association description, begging the answer to a fundamental point---so what? Even if it is true, is it a big deal? Would it answer an important clinical or biological question?
- 3) The claim of "local 3UTR hypermethylation was associated with the overexpression of VIRMA" is all about apparent associations with no causal evidence and investigations. Any molecular evidence that VIRMA binds to those hypermethylation sites and this binding is directly responsible for the change of m6A signal and then cancer progression?
- 4) Several places in the manuscript presented claims with no data support, for example, line 115-117, the author claimed that direct RNA-sequencing of the same batch of cells supports their work, but with no data presented to support this claim.

** As a service to authors, EMBO Press provides authors with the ability to transfer a manuscript that one journal cannot offer to publish to another journal, without the author having to upload the manuscript data again. To transfer your manuscript to another EMBO Press journal using this service, please click on Link Not Available

GERMAN
CANCER RESEARCH CENTER
IN THE HELMHOLTZ ASSOCIATION

German Cancer Research Center | M123 | PO Box 101949 | 69009 Heidelberg | Germany

Dr. Esther Schnapp
Editor
EMBO Reports

Division
Epigenetics
A130
Head:
Prof. Dr. Frank Lyko

Im Neuenheimer Feld 580
69120 Heidelberg
Germany
Phone +49 6221 42-3800
Telefax +49 6221 42-3802
f.lyko@dkfz.de
www.dkfz.de/epigenetics

Heidelberg, 25/09/2025

EMBOR-2025-62293V1

Dear Dr. Schnapp,

thank you for your note and for handling our manuscript. We appreciate the Referees' efforts and comments. I'm surprised, however, that you do not think that we can address the comments with a major revision and will explain this below:

Referee #1 highlights that we carefully produced and characterized a dataset of high technical quality that is a useful resource for the community. I would like to add that this dataset is the first to establish quantitative, transcriptome-wide m6A maps in any (!) human cancer and should therefore be of high general interest. We may have undersold this point in the submitted version of the manuscript and will be happy to more strongly emphasize it in a revised version.

The referee then states that it is difficult to draw firm conclusions regarding (1) the basis for the discrepancy between mass-spec and genomics, (2) a VIRMA-mediated regulon of hypermethylated stop codons leading to alternative polyadenylation, or (3) a link to UCB cancer progression. We think that we can improve the manuscript in all three areas as outlined below:

- 1) Our samples were obtained from the same source and with the same clinical protocol that was used for our previous study. We only changed the wording from "paratumoral" to "control" because our previous use of the term "paratumoral" has been considered ambiguous, particularly for samples obtained through cystectomies. We are sorry about the confusion and would be happy to clarify this point in the manuscript. Unfortunately, we cannot run parallel Masspec analyses with our mRNA preps, as it was already challenging to obtain sufficient amounts of tumor mRNA for GLORI-seq. However, we are prepared to analyze our GLORI-seq datasets for the abundance of non-mRNAs and provide the results as an additional quality control, as suggested by the reviewer.

Foundation under Public Law

Management Board
Prof. Dr. med. Michael Baumann
Ursula Weyrich

Deutsche Bank Heidelberg
IBAN: DE09 6727 0003 0015 7008 00
BIC (SWIFT): DEUT DES M672

Deutsche Bundesbank Karlsruhe
IBAN: DE39 6600 0000 0067 0019 02
BIC (SWIFT): MARK DEF 1660

- 2) We thank the reviewer for clearly delineating the issues regarding the proposed VIRMA-mediated regulon and for making specific suggestions about how to address them. We would appreciate having a chance to follow up on the reviewer's suggestions and challenge ("it would be upon the authors to firmly establish it") and are prepared to perform comprehensive additional analyses for a revised version.
- 3) While the manuscript was under consideration, we managed to successfully knock down VIRMA in two bladder cancer cell lines (one non-muscle invasive and one muscle invasive). Phenotypic analysis of these cells revealed reduced proliferation, apoptosis and colony formation in both models, which strongly suggests that VIRMA is a functionally relevant *bona fide* oncogene for UCB, consistent with our mutational analysis. These are exciting new findings that we would be happy to include in a revised version.

Referee #2 opens his/her evaluation by stating "This is an excellent study on m6A and cancer, using innovative, highly quantitative m6A mapping methods". All the referee's comments can be addressed with the available methods and datasets, and the results would further substantiate the rigor of our data analysis. We would be happy to include this in a revised version.

Referee #3's assessment is overall negative, but not particularly constructive. However, we expect that the inclusion of additional experiments and data analysis, as outlined above, will result in a more favorable impression.

In conclusion, we are confident that we can submit a thoroughly revised and substantially improved version in 3 months and would greatly appreciate the chance to do so.

With kind regards,

(Frank Lyko)

Dear Frank,

Thank you for your email asking us to reconsider our decision on your ms. I shared your rebuttal letter with referee 1, who is in principle open to re-assess a revised version of your ms, but also stresses that the revisions would have to be substantial:

Referee 1:

I've read the rebuttal. The fact that this study and the previous study relied on a similar dataset does mitigate that concern that I had raised, although it still lacks matching mass-spec and m6A samples, which limits the ability to make inferences regarding discrepancies. I was quite underwhelmed, though, by the evidence in support of the molecular roles ascribed to virma in the manuscript or the APA results. In both cases, this brief rebuttal does not present new data, but is simply asking for an opportunity to submit such new data. Based on this I don't feel I have a huge amount to work with. Should you decide to accept this manuscript for a revision, I'd be willing to read it once again and convey my thoughts, but it would have to be a quite substantial revision to be compelling.

I would thus like to invite you to revise your manuscript with the understanding that the referee concerns must be fully addressed and their suggestions taken on board. Please address all referee concerns in a complete point-by-point response. Acceptance of the manuscript will depend on a positive outcome of a second round of review. It is EMBO reports policy to allow a single round of major revision only and acceptance or rejection of the manuscript will therefore depend on the completeness of your responses included in the next, final version of the manuscript.

We realize that it is difficult to revise to a specific deadline. In the interest of protecting the conceptual advance provided by the work, we recommend a revision within 3 months (1st Jan 2026). Please discuss the revision progress ahead of this time with the editor if you require more time to complete the revisions.

- 1) A data availability section providing access to data deposited in public databases is missing. If you have not deposited any data, please add a sentence to the data availability section that explains that.
- 2) Your manuscript contains statistics and error bars based on $n=2$. Please use scatter blots in these cases. No statistics should be calculated if $n=2$.

5) a complete author checklist, which you can download from our author guidelines <<https://www.embopress.org/page/journal/14693178/authorguide>>. Please insert information in the checklist that is also reflected in the manuscript. The completed author checklist will also be part of the RPF.

6) Please note that all corresponding authors are required to supply an ORCID ID for their name upon submission of a revised manuscript (<<https://orcid.org/>>). Please find instructions on how to link your ORCID ID to your account in our manuscript tracking system in our Author guidelines <<https://www.embopress.org/page/journal/14693178/authorguide#authorshipguidelines>>

7) Before submitting your revision, primary datasets produced in this study need to be deposited in an appropriate public database (see <https://www.embopress.org/page/journal/14693178/authorguide#datadeposition>). Please remember to provide a reviewer password if the datasets are not yet public. The accession numbers and database should be listed in a formal "Data Availability" section placed after Materials & Method (see also <https://www.embopress.org/page/journal/14693178/authorguide#datadeposition>). Please note that the Data Availability Section is restricted to new primary data that are part of this study. * Note - All links should resolve to a page where the data can be accessed. *
If your study has not produced novel datasets, please mention this fact in the Data Availability Section.

- the name of the statistical test used to generate error bars and P values,
- the number (n) of independent experiments (please specify technical or biological replicates) underlying each data point,
- the nature of the bars and error bars (s.d., s.e.m.),
- If the data are obtained from $n < 3$, use scatter blots showing the individual data points.

12) All Materials and Methods need to be described in the main text using our 'Structured Methods' format, which is required for all research articles. According to this format, the Methods section includes a separate Reagents and Tools Table file (listing key reagents, experimental models, software and relevant equipment and including their sources and relevant identifiers) and a Methods and Protocols section describing the methods using a step-by-step protocol format. The aim is to facilitate adoption of the methodologies across labs. More information on how to adhere to this format as well as a downloadable template (.docx) for the Reagents and Tools Table can be found in our author guidelines: <https://www.embopress.org/page/journal/14693178/authorguide#structuredmethods>.

An example of a Method paper with Structured Methods can be found here: <https://www.embopress.org/doi/full/10.1038/s44320-024-00037-6#sec-4>

As part of the EMBO publication's Transparent Editorial Process, EMBO reports publishes online a Review Process File (RPF)

to accompany accepted manuscripts. This File will be published in conjunction with your paper and will include the referee reports, your point-by-point response and all pertinent correspondence relating to the manuscript.

I look forward to seeing a revised form of your manuscript when it is ready.

Koch et al., response to referee comments**Referee #1**

1. 'm6A dilution model' - in previous work the authors found substantial decreases in m6A levels between tumor and paratumor tissues, with m6A levels being reduced on average more than 50%. Here the authors compare tissues between cancer patients and controls, and find no differences between the two based on GLORI-based readouts. The authors view this as a contradiction, that they seek to resolve by weighting m6A stoichiometries by the expression levels of genes. There are multiple issues with this line of thought, and with the analyses. First, it is not even clear to use that there is any discrepancy between the results. A comparison of tissues to paratissues (previous paper) is very different from a comparison of cancerous tissue from one patient to healthy tissue from another. Perhaps there is simply no difference between the latter (that could be reproducibly detected based on both techniques), but there are considerable differences between the former. Second, given the centrality of this point, a crucial element this manuscript lacks is mass-spec measurements of m6A levels, to match the GLORI data, to allow evaluating whether truly there is a discrepancy. Third, the current conclusion is based on looking at the average weighted methylation levels across groups which qualitatively (but not quantitatively) shows a similar trend as observed in the past study (for other tissues). A much more convincing analysis, demonstrating that discrepancy between mass-spec studies and genomic ones can be bypassed using the approach suggested by the authors, would be to measure m6A via mass-spec for each of the samples (point 2 above) and then to correlate this against the GLORI based index. This is crucial, because mass-spec and genomic data can fail to correlate also for an abundance of other reasons, many of which completely technical (e.g. failure to fully remove rRNA contaminants).

>>> Our samples were obtained from the same source and with the same clinical protocol that was used for our previous study and therefore represent paratumoral tissues. However, as our clinical samples were usually small and because the majority of the material needed to be used for pathology, we could not generate sufficient amounts of mRNA for GLORI-seq from paired tumor-paratumoral samples. For the same reason, it was also not possible to provide the material for parallel Masspec analyses. The identity of the control samples has now been clarified in the text as "independent paratumoral control samples" (l. 91, l. 150, l. 283, l.372). To confirm the quality of our mRNA preparations, we have now analyzed our GLORI-seq datasets of T24 cells for the abundance of different RNA classes. We find that the large majority are mRNAs, while rRNAs only accounted for <0.01 %. The results are shown in our new Table EV 3 and mentioned in the text (l. 121-3).

2. 3' UTR hypermethylation associated with APA: The authors observe hypermethylated sites in cancer, and associate them with APA. We find these associations very weak. The authors demonstrate a (mild) enrichment towards decreased 3' UTR length in cancer, and they

demonstrate that loss of METTL3 leads to a similar phenomenon. The authors do not establish that these changes in 3' UTR occur within the same genes. Moreover, all the analyses are presented based on average m6A levels, despite the fact that substantial differences in m6A levels are present across tumor and control samples (Fig. 4B). A much more convincing analysis would be to rank samples based on m6A levels at individual sites, and establish that APA levels correlate with this. We should further highlight that in making their point about APA, the authors rely on a previous study (Yue, 2018) which itself makes only cautious claims about the relationship between m6A methylation and polyadenylation. In our reading of the literature the link between m6A and polyadenylation is tenuous at best, and it would be upon the authors to firmly establish it.

>>> To better characterize the relationships between VIRMA, 3' UTR hypermethylation, APA and bladder cancer phenotypes, we knocked down VIRMA in two bladder cancer cell lines. Subsequently, we compared these cells to controls using cancer phenotypic assays, GLORI-seq and RNA-seq. Our results show that VIRMA knockdown resulted in (1) strongly reduced m6A methylation, particularly in 3'-UTRs, (2) transcript lengthening, and (3) strongly reduced cancer phenotypes (proliferation, apoptosis resistance and colony formation). These results are shown in our new Fig. 6 and provide convincing confirmation for our initial assumptions connecting VIRMA overexpression to 3'-UTR hypermethylation, APA and oncogenic phenotypes.

3. APA associated with VIRMA: Unfortunately we found these analyses unconvincing as well. First, it is confusing aspect is that the authors attribute the effect to VIRMA (rather than to the entirety of the methylation machinery), and yet in their experiments choose to perturb METTL3 (and even report an expected effect). Second, results are again based on average levels across tumors/controls, whereas they would be much more convincing if the full power of 18 different samples could be used to establish an actual correlation between VIRMA levels and APA. The Kaplan-Meier plots appear to arbitrarily assign 345 samples to one group and 65 to another, and do not give the sense of being very robust.

>>> See point 2 above. Concerning the Kaplan-Meier analysis, we stratified patients using a data-driven optimal cutoff method. All unique expression values were evaluated as potential thresholds, excluding those that produced groups with fewer than 15 patients to avoid extreme splits. For each feasible cutoff, we performed a log-rank test and identified the threshold that maximized the separation in survival (DOI: 10.2307/2532740). We updated the methods section for clarification (l. 522-5).

4. Definition of Hyper- and Hypomethylated Transcripts: Clarification is required on how transcripts were classified as hyper- or hypomethylated. The manuscript currently lacks details regarding thresholds used and whether the classification is based on single or multiple sites per transcript. It should also be explicitly stated whether these sites behave consistently across entire transcripts or individually.

>>> Transcripts for which we found increased methylation levels in at least one m6A site were defined as "hypermethylated". Transcripts for which we found decreased methylation

levels in at least one m6A site were defined as “hypomethylated”. Transcripts, for which we detected bidirectional methylation changes, were designated as “hyper -and hypomethylated”. This is now explained in the text (l. 182-4) and in the legend to Fig. 3C. Fig. 3C also shows that most sites behave consistently across entire transcripts. This is now also explicitly mentioned in the text (l. 187-9).

Panel 5E: what would the corresponding panel look like for transcripts with positive delta-PUI? Is this truly an enrichment? Or just reflection of how m6A is distributed across genes? >>> As requested by the referee, we have modified Fig. 5E to include a comparison for transcripts with positive delta-PUI. The difference is highly significant, which is consistent with the notion that shortened transcripts are more likely to be m6A-methylated than lengthened transcripts. This is now clarified in the text (l. 237-9).

6. Overall, the language in manuscript is strong, whereas links are weak and associative. Language must be moderated.

>>> Our new data provides important confirmation for many key claims. Nevertheless, we have carefully screened the manuscript and toned down the language.

7. Differential Gene Expression: The manuscript should clarify how log-fold changes (logFC) of values as extreme as 10 or 20 were derived, as these appear unusually high.

>>> We thank the reviewer for alerting us to this issue. Data re-analysis showed that extreme log-fold changes were completely derived from transcripts with extremely low expression levels in one group. Based on this, we have now included an additional filtering step that requires a transcript to have a read count of ≥ 10 reads in every sample, which greatly reduced extreme expression changes (see our revised Figure EV 6). This filter was also applied to all downstream analyses in Fig. 4 and it did not affect their outcome.

Referee #2

1. I have a suggestion about how the discussion of the m6A sites detected using GLORI is described. The author say they have 88,000 sites. But the number of sites that you get in this method is very much dependent upon the threshold that you place on the number for independent reads required to call a site and the number of nonconversions needed. Since there are a substantial number of non-conversions that just occur at baseline, sometimes these putative sites might just be part of the background noise. Multiple independent non-conversions is the most convincing when calling a site. So, the author should explain what their criteria are. Additionally, of those 88,000, they mentioned that they occasionally get noncanonical sites. They attributed these to background. But they only show one of these sites and that site looks very similar to DRACH, which makes me think that these are real m6A events. How many sites were completely unrelated to DRACH? A problem with the study is that there is no great control possible for the samples. There's no knockout cell line.

But there needs to be better ways to understand what the background rates are. For instance, what is the non-conversion rate at sites that are unrelated to DRACH? If the non-conversion rate is one percent at these sites, it informs us about what type of false positive rate you might eventually get with that 88,000 called sites. There may be better ways to establish background. The inhibitor is a great and useful control, but it is problematic because the drug does not completely inhibit the enzyme and you don't know if the remaining m6A sites are real or not. This is why other methods need to be considered for establishing the background rate.

>>> We have comprehensively addressed this point through a number of additional analyses. The results are shown in our new Figure EV 2 and convincingly illustrate the specificity of our methylation calling approach. The point is also clarified in the main text (l. 128-39).

2. The authors mentioned improvements to GLORI. Are these similar to the ones used for GLORI2? It is worth mentioning this newer method and how it compares to the authors changes. Additionally, as part of the quality control discussion, it's really important to discuss. The minimum read length threshold that was placed for reads that were analyzed, but also a histogram should be presented for the distribution of read length sizes that were obtained and the threshold for use in the study. This can help us understand how many fragments were too small to be used and how much fragmentation was obtained in general. The authors might want to compare that to the size distribution from the original study. This helps to demonstrate that the methods that were used improved the overall protocol.

>>> Our improvements are distinct from GLORI 2.0 and include the reduction of mRNA fragmentation time from 4 to 3 min, the end repair of the converted mRNA (i.e. 3'-end dephosphorylation and 5'-end phosphorylation), and the optimization of library purification via TBE-PAGE to increase the number of sequencing-competent molecules. We also used a lower minimum read length threshold (20) compared to the original publication. A histogram for the distribution of read length sizes that were obtained is shown below. As the coverage and methylation calling is largely dependent on the mapping (which is likely to discard short fragments), we did not include this result in the manuscript. However, we clarified our modifications to the protocol in the methods section (l. 394-7).

Read length distribution across T24 cell line sequencing replicates. Histogram depicting the distribution of sequencing read lengths and read counts for each replicate of the analyzed T24 cell line. Each color represents one replicate.

3. The introduction talks about 18 cancer samples. But it was unclear to me from the phrasing if it was 18 samples total, some of which were cancer some of which were control or if it was 18 paired cancer and normal tissues.

>>> Our clinical samples were usually small because the majority of the material needs to be used for pathology. As such, it was challenging to obtain sufficient amounts of mRNA for GLORI-seq and we could not generate paired tumor-normal samples. Our analysis is based on 9 bladder cancer samples and 9 independent paratumoral control samples, which is now clarified in the text. Also see Referee #1, comment 1.

4. The study impact could be enhanced if the abstract and possibly even the title brings up the exciting conceptual questions that were tested in the study especially the idea that the authors bring up about focal changes versus global changes. The authors are quite correct that so many papers argue for focal changes, but this study really shows broad changes that are not highly selective for specific transcripts. This should really be emphasized because it is a major conceptual advance. Currently the way the title is written, it sounds like the study may have be of interest only to people who care about bladder cancer.

>>> We have carefully modified the title and abstract to put our findings in a broader context.

5. The authors discuss about 42,000 shared sites. But that implies that some sites were not shared. However, it is my suspicion that the sites that were not shared simply did not have enough read coverage in one sample versus the other. However, one could read this text to think that there are sites that are unique to one sample or to the control tissue that are simply not found in the bladder cancer despite sufficient coverage. An alternative possibility is that there may be a single non-conversion event in one sample, but not in another and that non-conversion event is not biologically meaningful. However, this would come across as a site ghat is unique in one sample versus other. I think greater clarity and perhaps an assessment of the sites that are shared after considering sufficient depth and creating a minimal threshold of the number of non-conversions that are required for calling an m6A that would be very helpful. I think based on this, I think the authors will conclude that all sites are shared between the samples. If so, this should be stated.

>>>To assess whether differences in m6A site detection across samples could be explained by coverage or non-conversion, we compared coverage and methylation levels between shared and non-shared m6A sites in tumor and control samples. Shared sites (detected in all 9 samples) showed higher median coverage (tumors: 129 vs. 37.7 reads; controls: 87.3 vs. 31.3 reads, Figure EV 3A and C) and higher median methylation levels (tumors: 0.495 vs. 0.278; controls: 0.491 vs. 0.273, Figure EV 3B and D) compared to sites detected in fewer samples. These findings are consistent with the notion that strong signals (i.e. a combination of high coverage and high methylation) are more likely to be shared. This is now clarified in the text (l. 159-66).

6. The criteria that the authors bring up is a 10% change in stoichiometry. But this seems extremely small. The author should discuss if they think that this type of change is likely to be biologically meaningful. I doubt it, but the author should mention and discuss this. To me, it would be worthwhile to indicate how many sites differ by 20% or 30% or 40%. It's important to remove Sites that have insufficient coverage. For instance, if there are two different samples and they each have a coverage of 10 reads, and the ground truth is 10% stoichiometry, simple statistical variation could allow one sample to have one out of 10 non-conversions and the other with two out of 10 non-conversions but this could both refer to 10%. When the authors talk about the number of sites that vary by 10%, is there sufficient read depth to know that this isn't simply statistical noise? I would suggest that they use or at least discuss the rigid criteria needed to know that these 10% stoichiometry changes are in fact real and not statistical noise. The text should talk much more about the read threshold numbers that are needed in order to be confident that a change in the m6A level is likely to be real rather than potentially due to random noise. For different sites with different levels of coverage, the m6A level would have to change to different degrees in order for that change to be meaningful. For instance of the site has 100 reads of coverage, a 10% change would be quite impressive. But if there were only 10 reads, that 10% might just be noise.

>>> Our analysis is based on the combination of three cutoffs: Coverage (>15, taken from the original GLORI publication), within-sample variability ($P < 0.05$, taken from the original GLORI publication), and Methylation difference >10% (defined in this study). To determine the lower band of the differential methylation cutoff, we determined the SD of the methylation level of each DRAC site in both tissue groups. The vast majority of sites had a low <10% SD in both tissue groups. Additional analysis showed that the mean SD of all SDs for each tissue group was roughly 1% (see our new Table EV 5). Furthermore, the SD of 90% of all sites varied less than 6% across samples. This means that a methylation level difference of 10% robustly exceeds the variability, and thus provides additional justification for 10% cutoff. These points have now been clarified in the text (l. 176-9).

7. Related to the point above, in the examples that are shown, it seems that all the sites seem to be elevated in the m6A level. Is this the trend that is seen? That an mRNA tends to show that all the sites are elevated or all the sites are decreased? Or do some sites go up in some sites go down?

>>> We have expanded our analysis and the results are shown in our modified Fig. 3C. We found only few transcripts that have both hyper- and hypomethylated m6A sites and the majority of transcripts is either hypermethylated OR hypomethylated. This is now clarified in the text (l. 187-9).

8. The dilution concept was quite interesting and novel, and the results about this really cast doubt on whether researchers really should draw deep interpretations from mass spectrometry experiments. So this was a very interesting part of the study. But when I read the abstract, it wasn't clear to me that the dilution wasn't caused by transcripts that are normally methylated, but now are not methylated. I think it's not clear. only when you read

the text, you realize that these are from transcripts that are normally not methylated and the overall expression changes.

>>> The abstract has been rewritten to emphasize that the UCB global methylation level becomes diluted due to expression changes of transcripts.

9. There is some discussion that previous studies used antibody based methods to do mapping. It should be pointed out that this method was not designed to be quantitative. It was designed to be qualitative. However, people used it in a quantitative manner without doing the proper Spike in standards are not using correct methods like m6A-seq2 which is more quantitative.

>>> Clarified in the text as suggested (l. 79-81).

10. The m6A and APA discussion is somewhat complicated. Keep in mind that Darnell did a study which showed that there were some small but measurable effect of m6A and APA. But these were very rare transcripts. A recent study by Ries et al. NSMB also found rare examples of m6A depletion leading to shorter 3'UTR. The Moline study is somewhat difficult to interpret because the method also looks at RNA's that contain m6Am, not just m6A so it's difficult to draw complete conclusion from that study. I would recommend caution when making the claim that m6A has clear widespread effects on APA.

>>> We have now mentioned in the discussion that the previously published studies on m6A and APA were difficult to interpret (l. 345-8) and have cautioned the general tone. Also, our newly generated data (Fig. 6) suggests a more direct link through VIRMA. Finally, we now mention that further analysis will be required to elucidate the underlying mechanisms (l. 348-55).

Referee #3

1) The entire analysis of this work is based on very small differences (i.e. delta m6A change >10% and $p < 0.05$ line 156), and there is no convincing data to support that such small change of m6A signal has a biological impact.

>>> We would like to point out that 10 % methylation difference is our lower cutoff. We determined this cutoff by within-sample variability analyses shown in our new Table EV 5. It is also important to notice that m6A methylation is a binary modification on the molecular level. For example, if there is a 10 % methylation reduction for an mRNA that is present in 1,000 copies per cell, 100 of these mRNAs have completely lost the methylation mark.

2) The claim of "the global loss of m6A signal in UCB is due to increased expression of unmethylated transcripts and decreased expression of highly methylated transcripts" itself is a superficial and hypothetical association description, begging the answer to a fundamental point---so what? Even if it is true, is it a big deal? Would it answer an important clinical or biological question?

>>> We would like to point out that Referee #2 considers this finding a “major conceptual advance” (see Referee #2, point 4). We have now further clarified this point in the text to hopefully better illustrate our point.

3) The claim of "local 3UTR hypermethylation was associated with the overexpression of VIRMA" is all about apparent associations with no causal evidence and investigations. Any molecular evidence that VIRMA binds to those hypermethylation sites and this binding is directly responsible for the change of m6A signal and then cancer progression?

>>> To better characterize the relationships between VIRMA and 3' UTR hypermethylation, we have now knocked down VIRMA in two bladder cancer cell lines. Subsequently, we performed comparative GLORI-seq with control cells. The results show that VIRMA knockdown resulted in strongly reduced 3' UTR hypermethylation. Also, the oncogenic phenotype of the bladder cancer cells was strongly reduced in terms of reduced proliferation and colony formation, as well as elevated Caspase 3/7 activity, indicating increased apoptosis signaling. These results are shown in our new Fig. 6 and Figure EV 8.

4) Several places in the manuscript presented claims with no data support, for example, line 115-117, the author claimed that direct RNA-sequencing of the same batch of cells supports their work, but with no data presented to support this claim.

>>> The dataset was cited as a biorxiv preprint in the initial submission. It has now been published and the corresponding reference (Hewel et al., 2025) has been updated.

Dear Frank,

Thank you for the submission of your revised manuscript and for your proposed revision plan that addresses the remaining referee concerns on your revised ms, as pasted below. Please go ahead and revise your ms one more time along the lines you and the referees suggest. Please also co-submit a point-by-point response with your final ms. If all cancer data are based on bladder cancer, this should be mentioned in the ms title.

A few editorial requests will also need to be addressed before we can proceed with the official acceptance of your manuscript:

- The author credits need to be removed from the ms file. All credits are entered during online ms submission.
- Please also add all funding info during online ms submission in the respective boxes.
- Please upload all EV tables as individual EV Table files (Table EV1-5). Each table legend and/or title needs to be part of the respective excel file.
- The Materials and Methods section should include a separate Reagents and Tools Table file (listing key reagents, experimental models, software and relevant equipment and including their sources and relevant identifiers) and a Methods and Protocols section in which methods should be described using a step-by-step protocol format with bullet points. More information is available in our guide to authors online: <https://link.springer.com/journal/44319/submission-guidelines>
- Our data integrity analyst noted that we need higher resolution source data for Blot images in Figures 6 and EV8. The reduction in resolution is commonly caused by converting original 16-bit TIFF files to RGB format for publication. Please supply source data at originally captured settings.

* Figure Legends - Comment s*

- Please note that the exact p values are not provided in the legends of figures 4B, 5F, G; 6F, G, H; EV8 F-H. Please provide exact values as reasonable.
- Please indicate the statistical test used for data analysis in the legends of figures 3A, 6C, EV8 C
- Please note that the box plots need to be defined in terms of minima, maxima, centre, bounds of box and whiskers, and percentile in the legends of figures 1C, 2C, 5F, G; EV2 D; EV3 A-D
- Please note that information related to n is missing in the legends of figures 1C, 2C, 3A, 4C, 5C, D, F; EV2 D, EV3 A-D

I would like to suggest some minor changes to the abstract that needs to be written in present tense. Please let me know whether you agree with this:

N6-Methyladenosine (m6A) is the most abundant internal modification of eukaryotic mRNAs and regulates target transcripts throughout the mRNA life cycle. Although changes in m6A have been reported in human cancers, technical limitations have hindered a comprehensive understanding of the cancer-associated m6A landscape. Here, we use GLORI-sequencing to establish the first transcriptome-wide, single-nucleotide resolution maps of m6A in bladder cancer. Comparing bladder cancer and healthy bladder samples, we discover two key m6A signatures: a global dilution of methylation and a focal hypermethylation at 3'-UTRs. The global methylation dilution results from an increased expression of unmethylated transcripts and a decreased expression of methylated transcripts. In contrast, focal 3'-UTR hypermethylation is associated with the overexpression of VIRMA, a component of the m6A writer complex. A functional role of VIRMA is confirmed in knockdown experiments that reveal reduced 3'-UTR methylation and oncogenic phenotypes of bladder cancer cells. Our study is the first to describe the m6A epitranscriptomic landscape of cancer at single-base resolution and provides first insights into the processes that generate its characteristic signatures.

EMBO press papers are accompanied online by A) a short (1-2 sentences) summary of the findings and their significance, B) 2-3 bullet points highlighting key results and C) a synopsis image that is exactly 550 pixels wide and 200-600 pixels high (the height is variable). The synopsis image should provide a sketch of the major findings, like a graphical abstract. Please note that text needs to be readable at the final size. Please send us this information along with the final manuscript.

Referee #1:

We had raised the concern that the fundamental observation at the center of this study - the gap between mass-spec and GLORI based measurements - was not sufficiently substantiated, given that the two techniques had not been applied to matched samples. While the authors explain that this reflects technical limitations, it does not set this concern at rest. As it stands, it remains unclear to this reviewer whether there truly is a gap between mass-spec based measurements and GLORI, with an alternative hypothesis being that differences arise from natural heterogeneity of samples.

Link between methylation and APA: The authors now show that they obtain a similar impact on APA upon depletion of VIRMA. This is important. Nonetheless, it remains unclear whether this link is direct or indirect. In my previous comments I had suggested that the authors assess the association between m6A levels per gene and the impact on APA per gene. The authors did not perform this critical analysis.

Bottom line: The manuscript has its strong suits: It puts together a large and high quality dataset of m6A in cancer and control cells, it observes a consistent link between depletion of the methylation machinery and APA, and it also observes indications for relevance of VIRMA in cancer. However, the manuscript still also suffers from substantial weaknesses: The key question that is raised (discrepancy between m6A measurements based on MS vs GLORI) is based on shaky ground, the answer is unconvincing, and how (whether at all?) this relates to APA through a direct mechanism remains to be established. Thus, the datasets and some of the observations here will help drive the field forward, and yet it is hard for me to formulate - on the basis of the current manuscript - a clear, conceptual take-home message that future studies will be able to build on.

I leave it to the discretion of the editor to decide whether this level of advance passes the bar of EMBO Reports.

Referee #2:

The authors have addressed all the key issues that have been raised. Overall, this study is very useful because it applies the latest, most powerful and accurate m6A mapping methods, and demonstrate the very specific and limited pattern of differences between normal and cancer tissues. The study shows a very different pattern of m6A changes than what is seen in other studies, and this study is likely to be much more accurate. They demonstrate that mass spectrometry can be deeply problematic for measuring or extrapolating m6A differences at specific sites. Overall, this will be an important and highly cited paper in the field.

Referee #3:

Thanks to the authors' efforts in trying to answer my questions, however, the key issues remain that 1) the claim of "the global loss of m6A signal in UCB is due to increased expression of unmethylated transcripts and decreased expression of highly methylated transcripts" itself is a superficial and hypothetical association description, not at all an important advance in science; 2) There is "no causal evidence and investigations" for the claim of "local 3'UTR hypermethylation was associated with the overexpression of VIRMA", no molecular evidence that VIRMA binds to those hypermethylation sites and this binding is directly responsible for the change of m6A signal and then cancer progression. I would not think this manuscript be suitable to be published at EMBO J..

Editor comments

1. As you will see, while referee 2 is positive, referees 1 and 3 still have concerns, and I would like to ask whether you can address referee 1's concern regarding the analysis of m6A levels and APA per single genes. Please let me know if more data can be provided.

>> Additional data has been provided, please see our response to Referee 1, point 2, below.

2. Apart from that, several conclusions need to be toned down throughout the ms and in the abstract. Please also mention bladder cancer in the title, as far as I understand this is the only cancer type investigated.

>> We re-added "bladder" to the title and accepted the suggested abstract into the manuscript.

Referee 1

1. We had raised the concern that the fundamental observation at the center of this study - the gap between mass-spec and GLORI based measurements - was not sufficiently substantiated, given that the two techniques had not been applied to matched samples. While the authors explain that this reflects technical limitations, it does not set this concern at rest. As it stands, it remains unclear to this reviewer whether there truly is a gap between mass-spec based measurements and GLORI, with an alternative hypothesis being that differences arise from natural heterogeneity of samples.

>> We consider this a peripheral question/finding/conclusion from our paper and have carefully checked the wording in the text to avoid any misunderstandings in this regard.

2. Link between methylation and APA: The authors now show that they obtain a similar impact on APA upon depletion of VIRMA. This is important. Nonetheless, it remains unclear whether this link is direct or indirect. In my previous comments I had suggested that the authors assess the association between m6A levels per gene and the impact on APA per gene. The authors did not perform this critical analysis.

>> We performed the analysis that was originally suggested by the Referee and did not observe a statistical correlation between $\Delta m6A$ and $\Delta PDUI$. However, it should be noted that both datasets (GLORI-seq and RNA-seq) were generated separately and by independent bulk methods. A direct analysis of the association between m6A levels per transcript and the impact on APA per transcript would, however, require the development of robust multi-modal single-molecule sequencing technologies that allow the simultaneous analysis of m6A levels and mRNA 3'-ends (discussed in I. 364-367). We have also identified specific transcripts that show both loss of methylation at the 3'-UTR and transcript lengthening upon VIRMA knockdown, which is now included in Figure EV9 (described in I. 277-283). Interestingly, differential m6A modification of some of these transcripts has been

prominently linked to bladder cancer in previous studies (e.g. ITGA6, PMID: 31409574 and AFF4, PMID: 30659266).

3. Bottom line: The manuscript has its strong suites: It puts together a large and high quality dataset of m6A in cancer and control cells, it observes a consistent link between depletion of the methylation machinery and APA, and it also observes indications for relevance of VIRMA in cancer. However, the manuscript still also suffers from substantial weaknesses: The key question that is raised (discrepancy between m6A measurements based on MS vs GLORI) is based on shaky ground, the answer is unconvincing, and how (whether at all?) this relates to APA through a direct mechanism remains to be established. Thus, the datasets and some of the observations here will help drive the field forward, and yet it is hard for me to formulate - on the basis of the current manuscript - a clear, conceptual take-home message that future studies will be able to build on.

>> We do not consider the discrepancy between m6A measurements based on MS vs GLORI a “key question” of the paper. It is only covered peripherally and the text has been carefully worded to reflect this. Also, we never claim that the discrepancy between m6A measurements based on MS vs GLORI is relevant for APA. Rather, we describe an independent mechanism (overexpression of VIRMA and focal 3'-UTR hypermethylation) that we link to APA. We agree that further work will be required to characterize the link between m6A and APA and have clarified this in the manuscript (l.344-359). However, we also believe that our data linking VIRMA to cancer-specific m6A patterning is both conceptually novel and supported by convincing data, both from clinical sources and from our cell-based models. This should not get lost in the APA discussion.

Referee 2

The authors have addressed all the key issues that have been raised. Overall, this study is very useful because it applies the latest, most powerful and accurate m6A mapping methods, and demonstrate the very specific and limited pattern of differences between normal and cancer tissues. The study shows a very different pattern of m6A changes than what is seen in other studies, and this study is likely to be much more accurate. They demonstrate that mass spectrometry can be deeply problematic for measuring or extrapolating m6A differences at specific sites. Overall, this will be a important and highly cited paper in the field.

>> We thank the Referee for his/her support.

Referee 3

1) The claim of "the global loss of m6A signal in UCB is due to increased expression of unmethylated transcripts and decreased expression of highly methylated transcripts" itself is

a superficial and hypothetical association description, not at all an important advance in science.

>> We respectfully disagree with the Referee's assessment. Our conclusion is not hypothetical but is directly supported by integrated experimental data: Using GLORI-seq and RNA-seq on patient samples, we observed widespread differential gene expression between bladder cancer and control tissues (Fig. EV6B), indicating that transcript abundance could influence global m6A measurements. We therefore performed an integrated analysis combining transcript-specific m6A methylation with transcript abundance, defining a weighted methylation level for each transcript. Summation of these weighted values revealed a global reduction of m6A in bladder cancer (Fig. 4B). Importantly, this effect was not primarily driven by a loss of m6A marks, but by the upregulation of abundant, unmethylated transcripts and the downregulation of abundant, highly methylated transcripts (Fig. 4C). Thus, the apparent hypomethylation reflects a transcriptome-wide compositional shift rather than a direct loss of methylation. To our knowledge, this is the first transcriptome-wide analysis integrating m6A methylation and expression to quantify global m6A levels. It is important because it provides a more informative measure than bulk assays that do not consider transcript expression.

2) There is "no causal evidence and investigations" for the claim of "local 3UTR hypermethylation was associated with the overexpression of VIRMA", no molecular evidence that VIRMA binds to those hypermethylation sites and this binding is directly responsible for the change of m6A signal and then cancer progression. I would not think this manuscript be suitable to be published at EMBO J.

>> In our VIRMA knockdowns, we observed that the strongest and densest methylation level reductions localized to regions surrounding the stop codon (Fig. 6D and Fig. EV8D) and state that this is consistent with a role of VIRMA in 3'-UTR methylation. We also observed that VIRMA knockdown strongly impaired cell proliferation and colony-forming capacity, accompanied by elevated Caspase-3/7 activity, indicating increased apoptotic signaling in both cell lines (Fig. 6F-H and Fig. EV8F-H). We state that these findings support a functional role for VIRMA in promoting tumor cell growth and survival. As such, we maintain that our conclusions are well substantiated by our experimental data.

Frank Lyko
German Cancer Research Center (DKFZ)
Epigenetics
Germany

Dear Frank,

I am very pleased to accept your manuscript for publication in the next available issue of EMBO reports. Thank you for your contribution to our journal.

You may qualify for financial assistance for your publication charges - either via a Springer Nature fully open access agreement or an EMBO initiative. Check your eligibility: <https://link.springer.com/journal/44319/how-to-publish-with-us>

>>> Please note that it is EMBO Reports policy for the transcript of the editorial process (containing referee reports and your response letter) to be published as an online supplement to each paper. If you do NOT want this, you will need to inform the Editorial Office via email immediately. More information is available here: <https://link.springer.com/partners/embo-press/editorial-policies#Peer%20review>